# Equilibrium Propagation for Non-Conservative Systems

**Antonino Emanuele Scurria** [1]   **Dimitri Vanden Abeele** [1]   **Bortolo Matteo Mognetti** [2]   **Serge Massar** [1]

## Abstract

Equilibrium Propagation (EP) is a physics-inspired learning algorithm that uses stationary states of a dynamical system both for inference and learning. In its original formulation it is limited to conservative systems, *i.e.* to dynamics which derive from an energy function. Given their applications, it is important to extend EP to non-conservative systems, *i.e.* systems with non-reciprocal interactions. Previous attempts to generalize EP to such systems failed to compute the exact gradient of the cost function. Here we propose a framework that extends EP to arbitrary non-conservative systems, including feedforward networks. We keep the key property of equilibrium propagation, namely the use of stationary states both for inference and learning. However, we modify the dynamics in the learning phase by a term proportional to the non-reciprocal part of the interaction so as to obtain the exact gradient of the cost function. This algorithm can also be derived using a variational formulation that generates the learning dynamics through an energy function defined over an augmented state space. Numerical experiments show that this algorithm achieves better performance and learns faster than previous proposals.

## 1. Introduction

Standard neural network optimization relies on error backpropagation, an algorithm whose computational mechanism is difficult to reconcile with biological (Crick, 1989) and physical implementations (Indiveri &Liu, 2015). Specifically, backpropagation requires a backward pass distinct from inference, the transmission of nonlocal error signals, and synchronous layer-wise computations with explicit gradient storage. These constraints have no clear analog in physical systems, making backpropagation challenging to implement in neuromorphic or analog hardware. Consequently, understanding how credit assignment can instead emerge from intrinsic system dynamics, through local interactions and continuous relaxation, is a central question in neuroscience and machine learning.

*Equilibrium Propagation* (EP) (Scellier &Bengio, 2017) represents one of the most promising advances in this direction. It formulates supervised learning as a contrast between two stationary states of a dynamical system: a 'free' phase where the system evolves autonomously, and a 'nudged' phase where outputs are weakly pushed toward their targets. The local change in neural states between these phases recovers the exact gradient of the cost function with respect to parameters. This enables spatially local learning exploiting the continuous relaxation of the system without a distinct backward circuit or explicit weight transport.

Since its introduction, several works have sought to improve the practicality and biological realism of EP. Algorithmic adaptations include enforcing temporal locality to avoid state storage (Ernoult et al., 2020; Falk et al., 2025), deriving agnostic updates for black-box energies (Scellier et al., 2022), and substituting nudging with clamping (Stern et al., 2021). Theoretically, the framework has been extended to stochastic systems (Scellier &Bengio, 2017; Massar &Mognetti, 2025) and Lagrangian dynamics for time-varying inputs (Massar, 2025; Pourcel et al., 2025; Berneman &Hexner, 2025). In parallel, simulations have explored suitable substrates, ranging from spiking (Martin et al., 2021; O'Connor et al., 2019) and resistive networks (Kendall et al., 2020) to coupled oscillators (Wang et al., 2024; Rageau &Grollier, 2025), as well as quantum systems (Wanjura &Marquardt, 2025; Massar &Mognetti, 2025; Scellier, 2024). Experimental realizations have been demonstrated in memristor crossbars (Yi et al., 2023), self-adjusting electrical circuits (Dillavou et al., 2022; 2024), elastic networks (Altman et al., 2024), and classical Ising models trained on quantum annealers (Laydevant et al., 2024).

Despite these recent developments and the theoretical el-

[1]Laboratoire d'Information Quantique (LIQ) CP224, Université libre de Bruxelles (ULB), Av. F. D. Roosevelt 50, 1050 Bruxelles, Belgium [2]Interdisciplinary Center for Nonlinear Phenomena and Complex Systems CP231, Université libre de Bruxelles (ULB), Av. F. D. Roosevelt 50, 1050 Bruxelles, Belgium. Correspondence to: Antonino Emanuele Scurria .

*Proceedings of the $43^{rd}$ International Conference on Machine Learning*, Seoul, South Korea. PMLR 306, 2026. Copyright 2026 by the author(s).

egance of EP, its standard formulation remains restricted to conservative systems. In these systems, dynamics are derived from an energy function, which inherently enforces symmetry (*e.g.*, symmetric synaptic connections $J_{ij} = J_{ji}$) through the action-reaction principle. This constraint precludes the use of EP in a broad class of models characterized by non-conservative forces. This includes the feedforward architectures dominant in modern AI, biological circuits, as well as physical systems that reach stationary states far from thermodynamic equilibrium, such as nonlinear optical systems driven by external lasers (Cin et al., 2025), opto-electronic systems (Kalinin et al., 2025), exciton-polariton condensates (Sajnok &Matuszewski, 2025), active meta-materials (Brandenbourger et al., 2019) and active colloids (Bishop et al., 2023; Osat &Golestanian, 2023) (see (Bowick et al., 2022) for a review).

Formally, we consider a dynamical system governed by a non-reciprocal force field $F(x, \theta, u)$, which relaxes to a stationary configuration $\overline{x}^0$ satisfying:

$$F(\overline{x}^0, \theta, u) = 0, \tag{1}$$

where $x$ represents the state variables, $\theta$ the learnable parameters and $u$ the static input. Our goal, given a target $y(u)$, is to compute the gradient of the cost function $C(\overline{x}^0, y)$ at this equilibrium,

$$\frac{\mathrm{d}C}{\mathrm{d}\theta}(\overline{x}^0, y), \tag{2}$$

and update $\theta$ to minimize the cost.

Previous attempts to extend EP to non-conservative dynamics include the *Vector Field* (VF) algorithm (Scellier et al., 2018). However, as noted by the authors, this method provides an unbiased gradient of the cost Eq. (2) only in the conservative case. To mitigate this, (Laborieux &Zenke, 2024) proposed adding a penalty to keep the Jacobian close to symmetry, essentially forcing the system to be as conservative as possible. Alternative methods related to VF, which similarly do not compute the exact gradient, were proposed in (Farinha et al., 2020; Costa &Santos, 2025) and for specific systems in simulation (Cin et al., 2025; Sajnok &Matuszewski, 2025).

Conversely, generalizations of backpropagation can handle non-reciprocal forces and compute the exact gradient of the cost Eq. (2) but inherit the same challenges in physical implementations. For instance, Backpropagation Through Time (Werbos, 1990) unfolds the network in time to apply standard backpropagation, Recurrent Backpropagation (Almeida, 1990; Pineda, 1987) avoids this memory requirement but still requires a specific circuit to propagate errors, and the continuous Adjoint Method (Chen et al., 2018) additionally requires integrating the dynamics backward in time which is not physically possible for a dissipative system.

In this paper, we first propose *Asymmetric EP* (AsymEP),

a framework where the original dynamics serve for inference, while a new augmented dynamic is used to compute gradients of the cost Eq. (2). In this augmented phase, the output neurons are nudged towards their targets (as in standard EP), while a local corrective term – proportional to the antisymmetric part of the Jacobian at the free equilibrium $\mathcal{J}_F(\overline{x}^0, \theta, u) = \frac{\partial}{\partial x}F(\overline{x}^0, \theta, u)$ – is added to the forces. The exact gradients of the cost with respect to parameters are then obtained by contrasting stationary states of the augmented system.

Second, we introduce *Dyadic EP*, a 'variational' approach to learning in non-conservative systems. This method involves doubling the number of variables in the system's state space and subsequently introducing a new energy function in this extended space. This approach takes advantage of the extended space to execute the positive and negative nudging phases in parallel, recovering the same computational cost as AsymEP. Derived from first principles, this approach is inspired by established methods for mapping dissipative dynamical systems onto conservative ones by doubling the degrees of freedom (Bateman, 1931; Galley, 2013; Aykroyd et al., 2025). A more comprehensive study of the theoretical framework and its application to feedforward networks can be found in (Scurria, 2026). Our method is related to the Dual Propagation algorithm (Høier et al., 2023; Høier &Zach, 2023; 2024) and constitutes an independent, first-principles generalization of Dyadic Learning (Nest &Høier; Høier et al., 2024)—previously limited to Hopfield networks—to arbitrary force fields.

Third, we validate our framework on MNIST (LeCun, 1998), Fashion-MNIST, and CIFAR-10. In continuous Hopfield networks initialized with symmetric connection matrices, AsymEP achieves better accuracy and learns faster than EP and VF. Additionally, when we constrain the network to have a strong degree of structural asymmetry, in which case EP is inapplicable, AsymEP outperforms VF. Finally, when we restrict connections to a feedforward structure, our algorithm effectively trains all parameters; in contrast, VF is limited to training the last layer, acting essentially as an Extreme Learning Machine (Huang et al., 2006; Wang et al., 2022) with poor performance.

In summary, this theoretical work proposes two generalizations of EP beyond conservative systems to arbitrary differentiable dynamics that compute in their stationary states.

## 2. Equilibrium Propagation Overview

### 2.1. Conservative Systems

We first review standard *Equilibrium Propagation* (EP) (Scellier &Bengio, 2017). We consider a network described by an energy function $E(x, \theta, u)$, such that the force field is

derived from the potential $E$:

$$F_E(x, \theta, u) = -\frac{\partial}{\partial x} E(x, \theta, u). \tag{3}$$

The objective is to compute the total gradient $\frac{dC}{d\theta}(\overline{x}^0, y)$ of a (quadratic) cost function $C(x, y)$ evaluated at the minimum energy configuration of the system. This *free equilibrium* denoted $\overline{x}^0$ (which depend implicitly in $\theta$ and $u$), satisfies the stationarity condition:

$$-\frac{\partial}{\partial x} E(\overline{x}^0, \theta, u) = 0. \tag{4}$$

To compute gradients, we introduce the augmented energy functional:

$$E_T(x, \theta, \beta, u, y) = E(x, \theta, u) + \beta C(x, y), \tag{5}$$

where $\beta$ is a scalar nudging parameter. The stationary configuration of this augmented system is obtained by integrating the dynamics

$$\frac{dx}{dt} = -\frac{\partial E_T(x, \theta, \beta, u)}{\partial x}, \tag{6}$$

until the energy minimum is reached. This new fixed point $\overline{x}^\beta$, called *nudged equilibrium*, satisfies:

$$\frac{\partial E(\overline{x}^\beta, \theta, u)}{\partial x} + \beta \frac{\partial C(\overline{x}^\beta, y)}{\partial x} = 0. \tag{7}$$

The training procedure, as improved in (Laborieux et al., 2021), uses two nudged phases with factors $\pm\beta$ (with $\beta \neq 0$). Starting from $\overline{x}^0$, the system relaxes to two nearby perturbed equilibria, $\overline{x}^{+\beta}$ and $\overline{x}^{-\beta}$. The displacement $\overline{x}^{+\beta} - \overline{x}^{-\beta}$ is then used to compute the parameter update in the learning rule:

$$\Delta\theta = -\epsilon \frac{1}{2\beta} \left( \frac{\partial E(\overline{x}^\beta, \theta, u)}{\partial \theta} - \frac{\partial E(\overline{x}^{-\beta}, \theta, u)}{\partial \theta} \right), \tag{8}$$

where $\epsilon > 0$ is the learning rate. The theoretical foundation of EP is the result that, in the $\lim_{\beta \to 0}$ of Eq. (8), we get:

$$\frac{dC(\overline{x}^0, y)}{d\theta} = \frac{d}{d\beta} \frac{\partial E(\overline{x}^\beta, \theta, u)}{\partial \theta}, \tag{9}$$

see Appendix D.1. The error of the above method is $O(\beta^2)$. This error can be further reduced using holomorphic equilibrium propagation (Laborieux &Zenke, 2022).

Thus, EP recovers the exact gradient of the cost function using only local computations. In this manner, learning implements gradient descent without an explicit backward pass, and credit assignment is realized through the system's intrinsic relaxation dynamics.

Three remarks can be made at this point. First, EP does not require the system to be at an energy minimum, but only at a stationary point, *i.e.*, that Eq. (7) holds. Second, EP implicitly assumes that the Jacobian $\mathcal{J}_E(\overline{x}^0, u) = \frac{\partial}{\partial x} F_E(\overline{x}^0, u)$ is invertible. In this work, we assume this condition holds and will not state it explicitly hereafter. Third, for simplicity, we omit the dependency on the input $u$ and target $y$ in the following equations.

### 2.2. Vector Field

The *Vector Field* (VF) algorithm, introduced in (Scellier et al., 2018), is an early attempt to adapt EP to non-reciprocal forces. This method relies on the observation that, for conservative systems, linearizing the right-hand side of Eq. (9) around the equilibrium point $\overline{x}^0$ yields

$$\lim_{\beta \to 0} \frac{1}{2\beta} \left( \frac{\partial E(\overline{x}^\beta, \theta)}{\partial \theta} - \frac{\partial E(\overline{x}^{-\beta}, \theta)}{\partial \theta} \right)$$
$$= \lim_{\beta \to 0} \left( -\frac{\partial F_E}{\partial \theta}(\overline{x}^0, \theta) \right)^\top \left( \frac{\overline{x}^\beta - \overline{x}^{-\beta}}{2\beta} \right), \tag{10}$$

where $F_E = -\partial_x E(x, \theta)$ is the conservative force. It is therefore tempting to use the right-hand side of Eq. (10) for parameter updates of non-conservative systems, for which no energy function $E$ exists.

The VF algorithm adopts precisely this approach. It uses the nudged counterpart of Eq. (7),

$$F(\overline{x}^\beta, \theta) - \beta \frac{\partial C}{\partial x}(\overline{x}^\beta) = 0, \tag{11}$$

in conjunction with the learning rule Eq. (10):

$$\Delta\theta = \epsilon \left( \frac{\partial F}{\partial \theta}(\overline{x}^0, \theta) \right)^\top \left( \frac{\overline{x}^\beta - \overline{x}^{-\beta}}{2\beta} \right). \tag{12}$$

However, as noted in (Scellier et al., 2018), Eq. (12) does not align with the true gradient $\frac{dC}{d\theta}(\overline{x}^0)$ and is exact only if the force is conservative. To see this, let $\mathcal{J}_F(x, \theta)$ denote the Jacobian of the vector field $F(x, \theta)$ (in components $(\mathcal{J}_F(x, \theta))_{ij} = \frac{\partial F_i(x, \theta)}{\partial x_j}$). Differentiating the equilibrium condition $F(\overline{x}^0, \theta) = 0$ with respect to $\theta$ gives

$$\mathcal{J}_F(\overline{x}^0, \theta) \frac{d\overline{x}^0}{d\theta} + \frac{\partial F}{\partial \theta}(\overline{x}^0, \theta) = 0. \tag{13}$$

Consequently, the exact gradient of the cost is

$$\frac{dC}{d\theta}(\overline{x}^0) = \frac{d\overline{x}^0}{d\theta}^\top \frac{\partial C}{\partial x}(\overline{x}^0)$$
$$= -\underbrace{\left( \frac{\partial F}{\partial \theta}(\overline{x}^0, \theta) \right)^\top}_{\text{pre-synaptic}} \underbrace{\left( (\mathcal{J}_F^\top(\overline{x}^0, \theta))^{-1} \frac{\partial C}{\partial x}(\overline{x}^0) \right)}_{\text{post-synaptic}}. \tag{14}$$

The terms 'pre-synaptic' and 'post-synaptic' in Eq. (14) are used by analogy with neuronal transmission: the pre-synaptic factor captures the local influence of $\theta$ on the force $F$, while the post-synaptic factor is the sensitivity of the cost to state perturbations.

If instead we differentiate the nudged equilibrium condition in Eq. (11) with respect to $\beta$ and evaluate at $\beta = 0$, we obtain

$$\mathcal{J}_F(\overline{x}^0, \theta) \frac{\mathrm{d}\overline{x}^\beta}{\mathrm{d}\beta}\bigg|_{\beta=0} - \frac{\partial C}{\partial x}(\overline{x}^0) = 0, \qquad (15)$$

which gives

$$\frac{\mathrm{d}\overline{x}^\beta}{\mathrm{d}\beta}\bigg|_{\beta=0} = \left(\mathcal{J}_F(\overline{x}^0, \theta)\right)^{-1} \frac{\partial C}{\partial x}(\overline{x}^0, y). \qquad (16)$$

The right-hand side of Eq. (16) represents the effective post-synaptic term used by the VF algorithm (Eq. 12). Comparing this with the exact post-synaptic term derived in Eq. (14), we see that they coincide only if $\mathcal{J}_F = \mathcal{J}_F^\top$, *i.e.*, only if the system is conservative.

Now, let $S_{\mathcal{J}}(\overline{x}^0, \theta)$ and $A_{\mathcal{J}}(\overline{x}^0, \theta)$ denote the symmetric and antisymmetric parts of the Jacobian at the free (un-nudged) equilibrium, respectively. Then, we show in Appendix A that the gradient error increases with the spectral radius of $\left(S_{\mathcal{J}}(\overline{x}^0, \theta)\right)^{-1} A_{\mathcal{J}}(\overline{x}^0, \theta)$. Consequently, large antisymmetric contributions degrade the gradient estimation, confirming empirical observations in the Appendix of (Ernoult et al., 2020). In fact, in the pathological limit where the Jacobian would be purely antisymmetric $S_{\mathcal{J}}(\overline{x}^0, \theta) = 0$, the update of VF gives the negative of the true gradient, maximizing the cost rather than minimizing it.

## 3. Asymmetric EP

Here, we introduce *Asymmetric EP* (AsymEP), see Algorithm 1, which removes the gradient estimate error inherent to VF by adding a local correction term to the augmented inference dynamics. The new nudged equilibrium $\overline{x}_A^\beta$ satisfies:

$$F(\overline{x}_A^\beta, \theta) - \beta\frac{\partial C}{\partial x}(\overline{x}_A^\beta) - 2A_{\mathcal{J}}(\overline{x}^0, \theta)(\overline{x}_A^\beta - \overline{x}^0) = 0, \quad (22)$$

As in VF, we then obtain two perturbed states $\overline{x}_A^{\pm\beta}$ for opposite nudging $\pm\beta$ and apply the contrastive learning rule of Eq. (12).

We now show that AsymEP gives rise to the correct learning rule, *i.e.* that right-hand side of Eq. (21) is proportional to the gradient of the cost function $\frac{\mathrm{d}C}{\mathrm{d}\theta}(\overline{x}^0)$ at the equilibrium point $\overline{x}^0$ (Eq. 14). To this end, note that the same reasoning leading to Eq. (16) leads to

$$\frac{\mathrm{d}\overline{x}_A^\beta}{\mathrm{d}\beta}\bigg|_{\beta=0} = \left(\mathcal{J}_{F_A}(\overline{x}^0, \theta)\right)^{-1} \frac{\partial C}{\partial x}(\overline{x}^0). \qquad (23)$$

---

**Algorithm 1** Asymmetric EP (AsymEP)

1: **Inputs:** Force field $F(x, \theta)$, cost function $C(x)$, nudging parameter $\beta$, learning rate $\epsilon$.
2: **repeat**
3:     **1. Free Phase: Evolve to stationary state**
4:         Evolve the system dynamics
5:

$$\frac{\mathrm{d}x}{\mathrm{d}t} = F(x, \theta), \qquad (17)$$

6:         until convergence to the stationary state $\overline{x}^0$.
7:     **2. Jacobian Decomposition**
8:         Compute the Jacobian at equilibrium:
9:

$$\mathcal{J}_F(\overline{x}^0, \theta) = \frac{\partial F}{\partial x}(\overline{x}^0, \theta), \qquad (18)$$

10:      and decompose it in its antisymmetric part:
11:

$$A_{\mathcal{J}}(\overline{x}^0, \theta) = \tfrac{1}{2}(\mathcal{J}_F(\overline{x}^0, \theta) - \mathcal{J}_F(\overline{x}^0, \theta)^\top). \quad (19)$$

12:     **3. Nudged Phase: Augmented Dynamics**
13:        Integrate the dynamics twice starting from $\overline{x}^0$
14:

$$\frac{\mathrm{d}x}{\mathrm{d}t} = F(x, \theta) - \beta\frac{\partial C}{\partial x}(x) - 2A_{\mathcal{J}}(\overline{x}^0, \theta)(x - \overline{x}^0),$$
$$(20)$$

15:      until convergence to two new stationary states $\overline{x}_A^{\pm\beta}$.
16: **4. Parameter Update**
17:      Update the parameters according to:
18:

$$\Delta\theta = \epsilon \left(\frac{\partial F}{\partial \theta}(\overline{x}^0, \theta)\right)^\top \left(\frac{\overline{x}_A^\beta - \overline{x}_A^{-\beta}}{2\beta}\right). \qquad (21)$$

19: **until** convergence of $\theta$
20: **Output:** Optimized parameters $\theta$.

---

where $\mathcal{J}_{F_A}(x, \theta)$ is the Jacobian of the modified dynamical system Eq. (20). At the equilibrium point $\overline{x}^0$, $\mathcal{J}_{F_A}$ is equal to the transpose of the original Jacobian:

$$\begin{aligned}
\mathcal{J}_{F_A}(\overline{x}^0, \theta) &= \mathcal{J}_F(\overline{x}^0, \theta) - 2A_{\mathcal{J}}(\overline{x}^0, \theta) \\
&= S_{\mathcal{J}}(\overline{x}^0, \theta) - A_{\mathcal{J}}(\overline{x}^0, \theta) \\
&= \mathcal{J}_F^\top(\overline{x}^0, \theta). \qquad (24)
\end{aligned}$$

where we have used the decomposition Eq. (44) of the original Jacobian $\mathcal{J}$ into its symmetric and antisymmetric components. Therefore, the left hand side of Eq. (23) is equal to the true post-synaptic term

$$\frac{\mathrm{d}\overline{x}_A^\beta}{\mathrm{d}\beta}\bigg|_{\beta=0} = \left(\mathcal{J}_F^\top(\overline{x}^0, \theta)\right)^{-1} \frac{\partial C}{\partial x}(\overline{x}^0), \qquad (25)$$

which, using Eq. (14), proves the result. Additionally, although implied by the equality with the true gradient, we explicitly demonstrate the equivalence of the gradient estimates obtained by AsymEP and Backpropagation Through Time in Appendix B following (Ernoult et al., 2019).

Note that the corrective term $-2A_{\mathcal{J}}(\overline{x}^0, \theta)(x - \overline{x}^0)$ in Eq. (20) is spatially local: $A_{\mathcal{J}}$ vanishes for unconnected neurons, and $(x - \overline{x}^0)$ is available at the synapse given the memory mechanism already required by Eq. (12). This correction can create backward connections (Section 5.3). However, in physical realizations, both feedforward and feedback connections must be physically present, though feedback may be deactivated during inference.

## 4. Dyadic EP

We now introduce *Dyadic EP* (Algorithm 2), a variational algorithm that computes the exact cost gradient in the limit of infinitesimal nudging. It maps the original $n$-variable dynamics $F(x, \theta)$ onto a $2n$-variable system $(z, z')$ defined by an energy $H(z, z', \theta)$ and cost $D(z, z')$. We show in Appendix E that AsymEP can be seen as the first-order projection of Dyadic EP onto the original $n$-dimensional state space.

The new system is defined by the energy $H$ and cost function $D$, given in terms of $F$ and $C$ by:

$$H(z, z', \theta) = -(z - z')^\top F\left(\frac{z + z'}{2}, \theta\right),$$
$$D(z, z') = C\left(\frac{z + z'}{2}\right), \quad (26)$$

where $z, z' \in \mathbb{R}^n$. In order to learn, we introduce the augmented energy

$$H_T(z, z', \theta, \beta) = H(z, z', \theta) + \beta D(z, z'). \quad (27)$$

The equilibrium configuration corresponds to a saddle point of $H_T$, where $z$ minimizes and $z'$ maximizes the energy. This poses no issue for EP, which requires only that the joint state $(z, z')$ reaches a stationary state. Although this min-maximization can be interpreted as $z$ evolving forward and $z'$ backward in time, in practice they evolve forward simultaneously, as we integrate the coupled equations:

$$\frac{dz}{dt} = -\frac{\partial H_T}{\partial z} = F\left(\frac{z + z'}{2}, \theta\right)$$
$$+ \left(\frac{z - z'}{2}\right)^\top \left.\frac{\partial F}{\partial z}\right|_{\frac{z+z'}{2}} - \frac{\beta}{2}\frac{\partial C}{\partial z}\left(\frac{z + z'}{2}\right),$$

$$\frac{dz'}{dt} = +\frac{\partial H_T}{\partial z'} = F\left(\frac{z + z'}{2}, \theta\right)$$
$$- \left(\frac{z - z'}{2}\right)^\top \left.\frac{\partial F}{\partial z'}\right|_{\frac{z+z'}{2}} + \frac{\beta}{2}\frac{\partial C}{\partial z'}\left(\frac{z + z'}{2}\right),$$
$$(28)$$

until a stationary point $(\overline{z}^\beta, \overline{z}'^\beta)$ is reached. Upon convergence, we follow the standard EP paradigm in using the difference $\overline{z}^\beta - \overline{z}'^\beta$ to compute the post-synaptic term. Under the change of variables $m = \frac{z+z'}{2}$ and $d = z - z'$, we prove in Appendix D that $m$ follows the original dynamics $F$ (ensuring valid inference), while $d$ relaxes to a "physical" error signal proportional to the cost gradient.

It is important to notice that while Dyadic EP introduces a distinct formulation, it remains consistent with the general theoretical setting of EP and matches the computational cost of AsymEP. Note also that we start the evolution of the free phase ($\beta = 0$) with the identical initial condition for $z$ and $z'$, (*i.e.*, $d = 0$). This guarantees that integrating Eq. (32) leads to a symmetric stationary point where $\overline{z}^0 = \overline{z}'^0$. Finally, we underline that the modified variational update rule in Eq. (34) is equivalent to the standard symmetric EP update rule in Eq. (8) (see Appendix D).

Now, to make this concrete, consider a continuous Hopfield network (see also Eq. (35)) with an asymmetric connection matrix $J$. After some calculations (see Appendix F), the augmented energy of the system can be re-expressed as:

$$H_T = -\frac{1}{2}\rho(z)^\top S\rho(z) + \frac{1}{2}\rho(z')^\top S\rho(z') - \rho(z)^\top A\rho(z')$$
$$+ \frac{1}{2}(\|z\|^2 - \|z'\|^2) + \frac{\beta}{2}\left(C(z, y) + C(z', y)\right), \quad (29)$$

where $S$ and $A$ are the symmetric and antisymmetric parts of $J$, respectively and $\rho$ is an element-wise non-linearity. An interesting analogy can be drawn with standard learning rules in discrete Hopfield networks (Hopfield, 1982). For a sequence of binary memories $\{\xi^1, \ldots, \xi^m\}$ where $\xi^\mu \in \{-1, 1\}^n$, $S$ corresponds to the standard autoassociative Hebbian rule $\sum_\mu \xi^\mu(\xi^\mu)^\top$, creating stable attractors at each pattern. In contrast, $A$ corresponds to the heteroassociative rule (*e.g.*, a cycle between $\xi^\mu$ and $\xi^\nu$ given by $\xi^\nu(\xi^\mu)^\top - \xi^\mu(\xi^\nu)^\top$), encoding transitions between patterns.

For this specific energy, the update rule given by Eq. (34) can be re-expressed as:

$$\Delta J \propto -\frac{1}{2\beta}\left(\rho(\overline{z}'^\beta) - \rho(\overline{z}^\beta)\right)\left[\rho(\overline{z}'^\beta) + \rho(\overline{z}^\beta)\right]^\top. \quad (30)$$

In the limit $\beta \to 0$, this gives:

$$\Delta J \propto \left(\frac{\overline{d}^\beta}{\beta}\right) \odot \rho'(\overline{m})\rho(\overline{m})^\top. \quad (31)$$

matching the learning rule in (Pineda, 1987), with $\lim_{\beta \to 0} \frac{\overline{d}^\beta}{\beta}$ being the error signal.

## 5. Numerical Experiments

In this section, we numerically validate AsymEP (Algorithm 1). The neuronal dynamics follows the one introduced

---

**Algorithm 2** Dyadic EP

---

1: **Inputs:** Force field $F(x, \theta)$, cost function $C(x, y)$, nudging parameter $\beta$, learning rate $\epsilon$
2: **repeat**
3:     **1. Free Phase: Evolve to stationary state**
4:         Evolve the system dynamics, starting from identical initial conditions $z(0) = z'(0) = z_0$,
5:
$$\frac{\mathrm{d}z}{\mathrm{d}t} = -\frac{\partial H}{\partial z}, \qquad \frac{\mathrm{d}z'}{\mathrm{d}t} = +\frac{\partial H}{\partial z'}, \qquad (32)$$
6:         until stationary states $\overline{z}^0, \overline{z}'^0$ are reached.
7:     **2. Nudged Equilibrium**
8:         Evolve the system dynamics, starting from the solution of the free phase $\overline{z}^0 = \overline{z}'^0$:
9:
$$\frac{\mathrm{d}z}{\mathrm{d}t} = -\frac{\partial H_T}{\partial z}, \qquad \frac{\mathrm{d}z'}{\mathrm{d}t} = +\frac{\partial H_T}{\partial z'}, \qquad (33)$$
10:         until two nudged stationary states $\overline{z}^\beta, \overline{z}'^\beta$ are reached.
11:     **3. Parameter Update**
12:         Update the parameters according to:
13:
$$\Delta\theta = -\epsilon \frac{1}{\beta} \left( \frac{\partial H(\overline{z}^\beta, \overline{z}'^\beta, \theta)}{\partial \theta} \right) \qquad (34)$$
14: **until** convergence of $\theta$
15: **Output:** Optimized parameters $\theta$.

---

in (Scellier &Bengio, 2017), and is generalized to allow for non-reciprocal forces as in (Scellier et al., 2018). For clarity, we express the forces in a form that explicitly separates the contributions of the external input and the recurrent interactions:

$$F(x) = \rho'(x) \odot \left( J^{\mathrm{in}} u + J^{\mathrm{dyn}} \rho(x) \right) - x, \qquad (35)$$

where $u \in \mathbb{R}^{N_{\mathrm{in}}}$ denotes the input and $x \in \mathbb{R}^{N_{\mathrm{dyn}}}$ the neuronal state, comprising both hidden and output units. The matrices $J^{\mathrm{in}} \in \mathbb{R}^{N_{\mathrm{dyn}} \times N_{\mathrm{in}}}$ and $J^{\mathrm{dyn}} \in \mathbb{R}^{N_{\mathrm{dyn}} \times N_{\mathrm{dyn}}}$ define the input and recurrent connectivity, respectively. The activation function $\rho(\cdot)$ is taken to be the hyperbolic tangent, applied element-wise.

If $J^{\mathrm{dyn}}$ is symmetric, we can define the energy:

$$E(x) = \frac{1}{2}\|x\|^2 - \frac{1}{2}\rho(x)^\top J^{\mathrm{dyn}}\rho(x) - \rho(x)^\top J^{\mathrm{in}} u, \quad (36)$$

which is identical to that of (Scellier &Bengio, 2017), provided that the input neurons are activated as $\rho(u)$.

Equation (35) naturally motivates a quantitative measure of structural asymmetry $r_{\mathrm{str}}$, defined as:

$$r_{\mathrm{str}} = \frac{\|(J^{\mathrm{dyn}\top} - J^{\mathrm{dyn}})/2\|_F}{\|J^{\mathrm{dyn}}\|_F}, \qquad (37)$$

where $\|\cdot\|_F$ denotes the Frobenius norm. Note that this metric does not capture the asymmetry of the Jacobian, which depends on the state $x$.

For numerical experiments, we restricted the network to a layered architecture with a single hidden layer to facilitate comparison with prior work. Accordingly, $J^{\mathrm{in}}$ contains only input-to-hidden connections, while $J^{\mathrm{dyn}}$ is block off-diagonal, encoding bidirectional interactions between the hidden and output layers. Both $J^{\mathrm{in}}$ and $J^{\mathrm{dyn}}$ are trained.

We first use MNIST (LeCun, 1998) (60k train, 10k test) followed by Fashion-MNIST to validate AsymEP, and then we further validate AsymEP and Dyadic EP by comparing them to Backpropagation on a convolutional feedforward, with CIFAR-10. Inputs are normalized using min-max to $[-1, 1]$ and targets are one-hot encoded in $\{-1, 1\}$. All hyperparameters are detailed in Appendix G, along with additional details and numerical results.

### 5.1. Symmetric Initialization

We start by comparing AsymEP with standard EP and VF. All algorithms are initialized with an identical symmetric matrix $J^{\mathrm{dyn}}$. EP maintains this symmetry throughout training, while VF and AsymEP induce asymmetry in $J^{\mathrm{dyn}}$. Since EP and VF already achieve strong performance on MNIST, the purpose of this experiment is to validate AsymEP and compare it against EP and VF rather than outperform the state of the art.

Figure 1 compares the three algorithms as a function of hidden-layer dimension after 1 and 20 epochs. AsymEP consistently outperforms the baselines, suggesting it learns faster and better.

Figure 2 studies the evolution of the asymmetry ratio $r_{\mathrm{str}}$. The results are reported for 50 hidden neurons. As expected, EP preserves the initial weight symmetry. In contrast, VF and AsymEP induce non-trivial evolution of $r_{\mathrm{str}}$ following two distinct patterns, resulting in three distinct network configurations. A complementary figure is available in Appendix G.1.

### 5.2. Fixed Asymmetry Ratio

While the previous section focused on networks compatible with all three algorithms (EP, VF, AsymEP), we now turn to architectures with strong structural asymmetry. In this regime, EP is inapplicable by construction, and, as we show, VF performs poorly, contrary to AsymEP which remains effective.

To this end, we consider a class of networks where the asymmetry ratio $r_{\mathrm{str}}$ defined in Eq. (37) is kept fixed. Let $\tilde{S}$ and $\tilde{A}$ be arbitrary symmetric and antisymmetric matrices in $\mathbb{R}^{N_{\mathrm{dyn}} \times N_{\mathrm{dyn}}}$ respectively. We enforce a fixed $r_{\mathrm{str}}$ via the

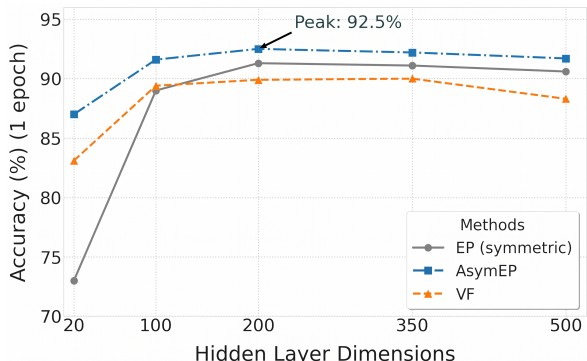

*(a)* Results after one epoch.

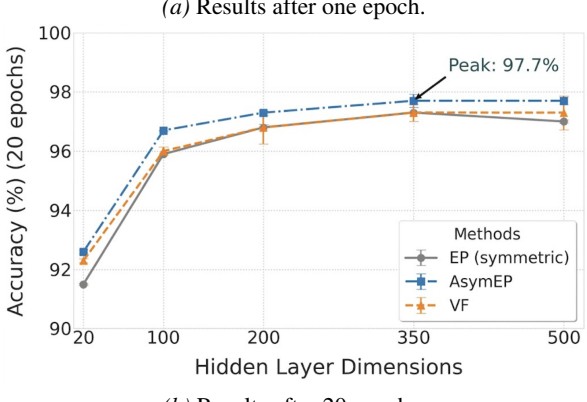

*(b)* Results after 20 epochs.

*Figure 1.* Comparison of algorithm performance on MNIST using a layered architecture with one hidden layer and symmetric initialization. Squares denote AsymEP, circles EP, and triangles VF. Test accuracy (averaged over 10 runs) is shown after one epoch (Fig. 1a) and 20 epochs (Fig. 1b).

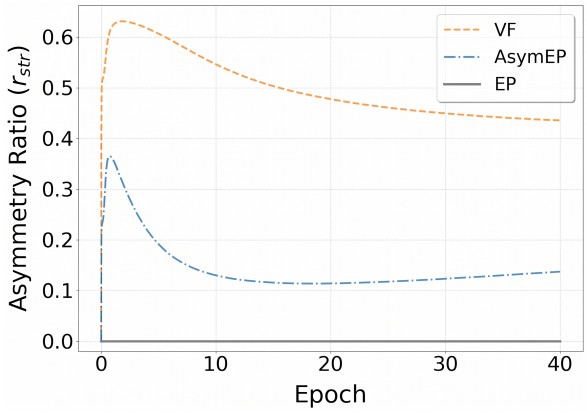

*Figure 2.* Evolution of the asymmetry ratio $r_{\text{str}}$ (defined in Eq. (37)) during training on MNIST for AsymEP, EP and VF, initialized from a symmetric configuration. The models use 50 hidden neurons.

following parameterization of the recurrent parameters:

$$J^{\text{dyn}} = \gamma \left[ \sqrt{1 - r_{\text{str}}^2} \frac{\tilde{S}}{\|\tilde{S}\|_F} + r_{\text{str}} \frac{\tilde{A}}{\|\tilde{A}\|_F} \right], \quad (38)$$

where $\gamma \in \mathbb{R}$ is a learnable global scale.

Using VF and AsymEP, we train a layered network with one hidden layer of 50 neurons (in which case $\tilde{S}$ and $\tilde{A}$ are block off-diagonal) for different values of $r_{\text{str}}$ to investigate the impact of structural asymmetry. We compare two training regimes: training only the input weights $J^{\text{in}}$ (and the scale $\gamma$), versus training all parameters including $J^{\text{dyn}}$. The first regime trains only the external forces from the input $\rho'(x) \odot J^{\text{in}} u$ (which correspond to a symmetric contribution in the Jacobian) applied to our non-conservative system, while the second additionally trains $J^{\text{dyn}}$ and therefore the non-symmetric part of the Jacobian directly.

Figure 3 summarizes the results. We find that AsymEP maintains robust performance across all asymmetry levels (*e.g.*, achieving an accuracy of $93.8 \pm 0.4\%$ at $r_{\text{str}} = 0$ and $94.9 \pm 0.2\%$ at $r_{\text{str}} = 0.875$ when training all parameters) and can even learn when the recurrent connection matrix $J^{\text{dyn}}$ is completely antisymmetric ($r_{\text{str}} = 1$). Additionally, training all parameters shows significant improvement over training only $J^{\text{in}}$.

In contrast, VF performs well at low asymmetry ratios but degrades as asymmetry increases, eventually dropping to chance levels (*e.g.*, accuracies of $5 \pm 3\%$ and $8 \pm 4\%$ at $r_{\text{str}} = 1$ for input-only and all-parameter training, respectively). When only $J^{\text{in}}$ is trained, VF accuracy collapses around $r_{\text{str}} \approx 0.5$, whereas training all parameters delays this collapse until $r_{\text{str}} \approx 0.8$. Our analysis in Appendix G.2.1 reveals that VF adjusts the dynamics such that the asymmetry of the Jacobian's off-diagonal terms remains strictly lower than the structural asymmetry ratio. The training appears to adjust the neuronal state such that neurons connected by strongly asymmetric weights have low activation. As shown in Appendix G.2.1, AsymEP learns faster than VF across all levels of asymmetry.

Finally, Appendix G.3 opens with a brief theoretical discussion of the stability of these non-conservative dynamics, followed by simulations on all-to-all topologies with constrained $r_{\text{str}}$ and input projections $J^{\text{in}}$. Even in this worst-case setting, AsymEP reduces oscillations and improves stability.

### 5.3. Feedforward Architectures

We now consider a purely feedforward architecture. Here VF trains only the last layer: with no backward connections, the output nudging signal cannot reach earlier layers, so for every layer but the last the nudged stationary states coincide with the free states, giving zero weight updates. As only the output layer is trained, the system essentially becomes an Extreme Learning Machine (Huang et al., 2006; Wang et al., 2022). In contrast, AsymEP introduces a correction that generates effective backward connections, allowing the

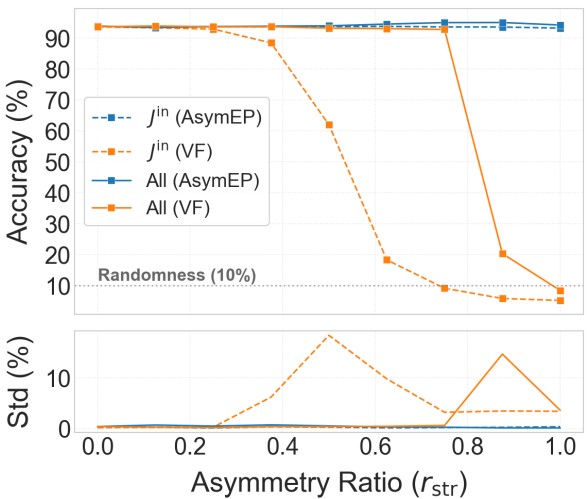

*Figure 3.* Impact of the structural asymmetry ratio $r_{str}$ on accuracy (top) and standard deviation over 10 runs (bottom) on MNIST. We compare VF (orange) and AsymEP (blue) under two training regimes: training only $J^{in}$ (dashed) or all parameters (solid).

nudging signal to influence all layers. We make this explicit for a network with one hidden layer.

Let the state $x$ be partitioned in hidden $h$ and output $o$ layers. The recurrent connection matrix is then $J^{dyn} = \begin{bmatrix} 0 & 0 \\ W_{h \to o} & 0 \end{bmatrix}$. The forces of the system are:

$$\begin{cases} F_h^\beta = \rho'(h) \odot \left( J^{in}u + \lambda(W_{h \to o})^\top(o - \bar{o}^0) \right) - h \\ F_o^\beta = \rho'(o) \odot \left( W_{h \to o}\rho(h) - \lambda W_{h \to o}(h - \bar{h}^0) \right) \\ \qquad + \lambda\beta\frac{\partial C}{\partial o} - o \end{cases} \quad (39)$$

where $\lambda$ is 0 during the free inference and 1 during the nudged phase (Eq. 20). The force on the hidden layer $F_h^\beta$ now depends on the output layer through the term $\rho'(h) \odot (W_{h \to o})^\top (o - \bar{o}^0)$, enabling the nudge (the term $\beta\frac{\partial C}{\partial o}$) to influence the hidden layer. This implicitly assumes that the hardware implementation supports the physical activation of these backward connections.

We validate this using a single hidden layer of only 20 neurons on MNIST. After training, VF saturates with $64.3 \pm 2.0\%$ accuracy, whereas AsymEP reaches $92.7 \pm 0.5\%$ accuracy. We expect this discrepancy to increase with network depth, since this increases the number of layers unable to learn under VF. A figure with the accuracy during training can be found in Appendix G.4.2.

### 5.4. Advantages of Non-Conservative Dynamics

AsymEP is not tied to a specific neural dynamics. To further assess the benefits of training non-conservative dynamics using AsymEP, we compare several dynamics and connec-

tivity structures inspired by (Millidge et al., 2023), while keeping the number of trainable parameters fixed.

Experiments are conducted on Fashion-MNIST using a two-hidden-layer network with hidden dimensions 500 and 200. Network states are denoted $(x_0, x_1, x_2, x_3)$, where $x_0$ is the input and $x_3 = x_L$ the output. Forward and backward connections are denoted by $W_k$ and $B_k$, respectively, with $W_1 = J^{in}$.

We consider three classes of dynamics. First, the Continuous Hopfield (CH) dynamics introduced previously:

$$\frac{dx_k}{dt} = -x_k + \rho'(x_k) \odot \left( W_k\rho(x_{k-1}) + (1 - \delta_{k,L})B_k\rho(x_{k+1}) \right). \quad (40)$$

Second, Predictive Coding (PC) dynamics, defined through the prediction errors $e_k = x_k - W_k\rho(x_{k-1})$, whose fixed point $e_k = 0$ corresponds to a standard feedforward network:

$$\frac{dx_k}{dt} = -e_k + (1 - \delta_{k,L})\left( \rho'(x_k) \odot (B_k e_{k+1}) \right). \quad (41)$$

Third, a standard dynamics chosen for direct comparison with backpropagation:

$$\frac{dx_k}{dt} = -x_k + W_k\rho(x_{k-1}) + (1 - \delta_{k,L})B_k\rho(x_{k+1}). \quad (42)$$

For each dynamics, we examine three connectivity scenarios.

- In the *asymmetric* case ($B_k \neq W_{k+1}^\top$), the backward weights $B_k$ are randomly initialized and kept fixed while only the forward weights are trained, ensuring a fair comparison (*i.e.*, identical number of parameters); in PC, the learning rule for $B_k$ is zero when only inputs are clamped.

- In the *symmetric / conservative* case ($B_k = W_{k+1}^\top$), the CH and PC dynamics derive from an energy functional, while the standard dynamics remains non-conservative due to its non-symmetric Jacobian.

- In the *feedforward* case ($B_k = 0$), the PC and standard dynamics coincide; for the standard dynamics, the AsymEP learning rule mirrors backpropagation, with $\Delta x_k^\beta = \frac{1}{2\beta}(x^\beta - x^{-\beta})$ acting as the propagated error signal.

Table 1 shows that AsymEP consistently outperforms VF in both asymmetric and feedforward settings, in final accuracy, learning speed, and stability. After a single epoch it already provides on average a 15% accuracy gain with an order-of-magnitude reduction in variance. Remarkably, AsymEP with asymmetric connectivity also surpasses EP on symmetric networks despite training only the forward

weights, suggesting that relaxing symmetry constraints may improve expressivity. Supplementary results are provided in Appendix G.5.

*Table 1.* Test accuracy on Fashion-MNIST (%) at Epoch 50 (mean $\pm$ std 10 runs). BP on a standard feedforward architecture using MSE and SGD achieve $87.37 \pm 0.29\%$.

|  |  | EP | AsymEP | VF |
|---|---|---|---|---|
| CH | Asym | - | $86.78 \pm 0.14$ | $85.20 \pm 0.12$ |
|  | Feedfor | - | $86.05 \pm 0.12$ | $77.76 \pm 0.37$ |
|  | Sym | $84.30 \pm 0.13$ | - | - |
| PC | Asym | - | $86.20 \pm 0.17$ | $80.71 \pm 6.17$ |
|  | Sym | $84.78 \pm 0.14$ | - | - |
| Standard | Asym | - | $82.91 \pm 0.48$ | $75.52 \pm 1.69$ |
|  | Feedfor | - | $86.25 \pm 0.16$ | $78.58 \pm 0.28$ |

Finally, to investigate how AsymEP scales with depth, we trained deeper fully connected networks with two and three hidden layers of 500 neurons on Fashion-MNIST, reaching $86.41 \pm 0.22\%$ and $87.8 \pm 0.15\%$ test accuracy respectively.

### 5.5. Feedforward Training on CIFAR-10: BP vs. Dyadic EP vs. AsymEP

To test whether our framework scales beyond shallow networks, we conclude with a deep, purely feedforward CNN architecture trained on CIFAR-10. We compare backpropagation (BP), VF, AsymEP and Dyadic EP in a controlled setting where the gradient estimator is the only difference between runs: all methods share the same configuration, with the BP gradient replaced by the contrast of stationary states for the EP-based methods (see App. G.6 for details). Each configuration is trained for 40 epochs over 5 seeds.

Table 2 reports the final test accuracy. Both of our algorithms scale to this regime, closely tracking the BP baseline throughout training and matching its final accuracy: a paired $t$-test finds no significant difference between Dyadic EP and BP ($p = 0.75$), and only a sub-percent gap for AsymEP. In contrast, VF makes slight initial progress (peaking near $30\%$) before collapsing to chance level ($10\%$). Additional details can be found in Appendix G.6

*Table 2.* Test accuracy on CIFAR-10 (%) at epoch 40 (mean $\pm$ std over 5 seeds).

| Method | Test Acc. (%) |
|---|---|
| Backpropagation | $90.66 \pm 0.25$ |
| Dyadic EP | $90.69 \pm 0.14$ |
| AsymEP | $89.74 \pm 0.14$ |
| VF | $10.00 \pm 0.00$ |

## 6. Discussion and Conclusion

In this work, we extended Equilibrium Propagation (EP) to non-conservative systems that reach stationary states by deriving two mathematically equivalent algorithms that recover the exact gradient of the cost function in the limit of infinitesimal nudging.

The first approach, *Asymmetric EP*, preserves the original inference dynamics. It introduces a corrective force during the nudged phase that remains spatially local, as the antisymmetric Jacobian is null for unconnected neurons and the perturbation from equilibrium is available at the synapse level. Unlike standard methods like Recurrent Backpropagation (Almeida, 1990; Pineda, 1987), this avoids explicit digital weight transposition. However, a physical mechanism to obtain the local corrective force at the synapse level remains a subject for future work. We also note that AsymEP shares the temporal non-locality of standard EP.

The second approach, *Dyadic EP*, doubles the state space to map non-reciprocal dynamics onto an energy landscape—conceptually reminiscent of multi-compartment cortical neurons, where apical dendrites integrate feedback (analogous to $z - z'$) separately from basal feedforward input (analogous to $z + z'$) (Guerguiev et al., 2017). Additionally, this expanded space also enables the positive and negative nudging phases to run in parallel. This offers a pathway to implement a version of EP that is local in time, but would require a doubling of the degrees of freedom on the physical hardware. More fundamentally, the energy defined on the extended state shows that the tools and theoretical guarantees obtained for EP should also apply to the case of non-reciprocal forces, and that the variational principle behind EP is universal in the sense that it can be applied to all networks which operate in a stationary state.

Furthermore, Dyadic EP is not restricted to the EP community and could suggest a more physically plausible alternative to the stationary-state Adjoint Method (for fixed inputs) (Chen et al., 2018): by solving the forward and adjoint equations simultaneously via relaxation, it circumvents a separate backward-in-time pass.

Finally, our experiments on MNIST, Fashion-MNIST, and CIFAR-10 confirm that AsymEP and Dyadic EP consistently outperform EP and VF, and notably enables effective training of feedforward networks.

Our work thus opens new avenues for learning in neuromorphic hardware, dissipative physical systems, and neural architectures where asymmetry is intrinsic rather than incidental.

## Impact Statement

This paper presents results that advance the field of machine learning. There are many potential societal consequences of our work, none of which we feel must be specifically highlighted here.

## Acknowledgments

AES is fully funded by the Horizon Europe Marie Skłodowska-Curie Doctoral Network 'Postdigital Plus' (Grant 101169118). DVA acknowledges the support of the French Community of Belgium through a FRIA fellowship. SM acknowledges financial support by the Fonds de la Recherche Scientifique–FNRS, Belgium under EOS Project No. 40007536. Computational resources have been provided by the Consortium des Équipements de Calcul Intensif (CÉCI), funded by the Fonds de la Recherche Scientifique de Belgique (F.R.S.-FNRS) under Grant No. 2.5020.11 and by the Walloon Region.

*"ἁρμονίη ἀφανὴς φανερῆς κρείττων"*

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

## A. Gradient Estimation Error in VF

In this appendix, we quantify the gradient estimation error introduced by VF in the limit where the Jacobian asymmetry is small.

Comparing the post-synaptic update terms in Eqs. (12) and (14) gives the following error in the gradient of the cost:

$$\text{Error} = -\left(\frac{\partial F}{\partial \theta}(\overline{x}^0, \theta)\right)^\top$$
$$\times \left(\left(\mathcal{J}_F(\overline{x}^0, \theta)\right)^{-1} - \left(\mathcal{J}_F^\top(\overline{x}^0, \theta)\right)^{-1}\right)\frac{\partial C}{\partial x}(\overline{x}^0, y), \quad (43)$$

To quantify this error, we decompose the Jacobian $\mathcal{J}_F(x, \theta)$ into its symmetric part $S_\mathcal{J}(x, \theta)$ and antisymmetric part

$$S_\mathcal{J}(x, \theta) = \tfrac{1}{2}\left(\mathcal{J}_F(x, \theta) + \mathcal{J}_F^\top(x, \theta)\right),$$
$$A_\mathcal{J}(x, \theta) = \tfrac{1}{2}\left(\mathcal{J}_F(x, \theta) - \mathcal{J}_F^\top(x, \theta)\right). \quad (44)$$

Assuming the asymmetry $A_\mathcal{J}(x, \theta)$ is small, we can make a series expansion in $S_\mathcal{J}^{-1}A_\mathcal{J}$ (omitting the dependencies for clarity). Applying the Neumann expansion for small $\|S_\mathcal{J}^{-1}A_\mathcal{J}\|$ gives

$$(\mathcal{J}_F)^{-1} = \left(\sum_{n=0}^{\infty}(-1)^n(S_\mathcal{J}^{-1}A_\mathcal{J})^n\right)S_\mathcal{J}^{-1}, \quad (45)$$

$$(\mathcal{J}_F^\top)^{-1} = \left(\sum_{n=0}^{\infty}(S_\mathcal{J}^{-1}A_\mathcal{J})^n\right)S_\mathcal{J}^{-1}. \quad (46)$$

Subtracting the two series and assuming convergence, we finally obtain

$$(\mathcal{J}_F)^{-1} - (\mathcal{J}_F^\top)^{-1} = -2\left(\sum_{n=0}^{\infty}\left(S_\mathcal{J}^{-1}A_\mathcal{J}\right)^{2n+1}\right)S_\mathcal{J}^{-1}. \quad (47)$$

## B. Equivalence between AsymEP and BPTT

In this appendix, we sketch the equivalence between the gradient estimate computed by AsymEP and Backpropagation Through Time (BPTT) (Werbos, 1990) for a Recurrent Neural Network with fixed inputs. Our derivation relies on the proof provided by Ernoult et al. (2019), which established that standard (conservative) EP computes gradients identical to those of BPTT. To facilitate direct comparison, we adopt their notation for this section.

The proof provided by Ernoult et al. (2019) relies on the assumption that the vector field $F$ (*i.e.*, transition function) is derived from a scalar potential function, which implies that

$$\frac{\partial F}{\partial s} = \left(\frac{\partial F}{\partial s}\right)^\top, \quad (48)$$

where $s$ denotes the dynamical state of the system. This symmetry is the linchpin of the equivalence proof, as the gradient expressions derived for BPTT and standard EP differ precisely by a transpose operation applied to $\frac{\partial F}{\partial s}$.

This observation aligns with our analysis in the main text: VF fails in non-conservative systems due to the missing transpose in the post-synaptic term (see Eq. (16)). Following the derivation in Ernoult et al. (2019) (*viz.*, Appendix A, Eqs. (31–33)), the recursive relations for the gradients in BPTT are given by:

$$\nabla_s^{\text{BPTT}}(0) = \frac{\partial \ell}{\partial s}(s_\star, y), \quad (49)$$

and for all $t = 1, \ldots, K$,

$$\nabla_s^{\text{BPTT}}(t) = \left(\frac{\partial F}{\partial s}(x, s_\star, \theta)\right)^\top \nabla_s^{\text{BPTT}}(t-1), \quad (50)$$

$$\nabla_\theta^{\text{BPTT}}(t) = \left(\frac{\partial F}{\partial \theta}(x, s_\star, \theta)\right)^\top \nabla_s^{\text{BPTT}}(t-1), \quad (51)$$

where $\theta$ represents the optimization parameters, $\ell$ is the cost function, $s_\star$ is the free equilibrium state (satisfying $F(s_\star) = 0$), $y$ is the target, and $x$ is the input. The index $t$ denotes the unrolled time steps, initialized at $s(0) = s_\star$.

In contrast, the gradients computed by VF follow the recursion (*viz.*, Ernoult et al. (2019), Appendix A, Eqs. (24–26)):

$$\Delta_s^{\text{EP}}(0) = -\frac{\partial \ell}{\partial s}(s_\star, y), \quad (52)$$

and for all $t \geq 0$,

$$\Delta_s^{\text{EP}}(t+1) = \frac{\partial F}{\partial s}(x, s_\star, \theta)\,\Delta_s^{\text{EP}}(t), \quad (53)$$

$$\Delta_\theta^{\text{EP}}(t+1) = \left(\frac{\partial F}{\partial \theta}(x, s_\star, \theta)\right)^\top \Delta_s^{\text{EP}}(t). \quad (54)$$

Comparing these two sets of equations confirms that the only difference are Eqs. (50) and (53), specifically the transpose of the Jacobian $\frac{\partial F}{\partial s}$ (ignoring the global sign difference in Eqs. (49) and (52)).

In AsymEP, we modify the dynamics by adding a correction term dependent on the antisymmetric part of the Jacobian. Denoting the force of this augmented system by $F^A$, the Jacobian at the free equilibrium satisfies:

$$\frac{\partial F^A}{\partial s}(x, s_\star, \theta) = \left(\frac{\partial F}{\partial s}(x, s_\star, \theta)\right)^\top. \quad (55)$$

By substituting this corrected Jacobian into the recursive relations, AsymEP recovers the exact transpose required by BPTT. Consequently, our method extends the equivalence between EP and BPTT to the general case of non-conservative force.

## C. Out-of-Equilibrium Mechanics

Here we sketch the physical picture behind the doubled-energy construction of Eq. (26). The full derivation from Hamilton's least-action principle, together with its connection to the Bateman–Galley formalism for non-conservative classical mechanics (Bateman, 1931; Galley, 2013; Aykroyd et al., 2025), can be found in (Scurria, 2026).

### C.1. The Helmholtz Obstruction

The natural physical route to a variational principle for a dynamical system $\dot{x} = F(x, \theta)$ is to seek a scalar potential $E$ such that $F = -\partial_x E$. The classical Helmholtz integrability condition states that such an $E$ exists if and only if the Jacobian $\mathcal{J}_F$ is symmetric everywhere. Whenever the interactions are non-reciprocal — as in feedforward networks, active matter, or driven optical systems — $\mathcal{J}_F$ acquires a non-zero antisymmetric part and the Helmholtz condition fails identically. No scalar potential on the original $n$-dimensional state space can then generate the dynamics, and the "energy minimisation" route at the heart of standard EP is blocked at the structural level. The obstruction is not a matter of computational convenience: it reflects the fact that the rotational component of $F$ carries information that no scalar function of $x$ alone can record.

### C.2. Variational Reconstruction on a Doubled Space

Applying the Bateman–Galley formalism circumvents this obstruction by enlarging the configuration space. The single state $x \in \mathbb{R}^n$ is replaced by a conjugate pair $(z, z') \in \mathbb{R}^{2n}$, and the rotational component of $F$ — which has no scalar generator on the original $n$-dimensional space — is absorbed into a bilinear coupling between $z$ and $z'$ on the doubled space, where it does admit a variational description. The physical motion is recovered on the diagonal submanifold $z = z'$ (the so called 'physical limit'), while the off-diagonal direction $d = z - z'$ supplies the additional degree of freedom needed to encode non-reciprocity.

Specializing this reconstruction to the overdamped (first-order) regime relevant to relaxational neural dynamics yields the bilinear energy

$$H(z, z', \theta) = -(z - z')^\top F\left(\frac{z + z'}{2}, \theta\right), \quad (56)$$

which is precisely Eq. (26). The symmetric midpoint $m = (z + z')/2$ plays the role of the physical coordinate of the doubled system, while $d$ is the auxiliary direction along which non-reciprocity is stored. On the submanifold $z = z'$ the coupling proportional to $(z - z')$ vanishes identically and both states evolve under the original field $F$, so the doubling leaves the on-shell physics unchanged. We refer the reader to (Scurria, 2026) for the full construction.

### C.3. Symmetry Breaking as Credit Assignment

On the diagonal manifold $z = z'$ the doubled system enjoys a gauge symmetry: the auxiliary variable $z'$ is redundant and the difference $d$ is identically zero. Credit assignment is implemented by deliberately breaking this symmetry through the task cost. Adding $\beta D(z, z') = \beta C(m)$ to $H$ exerts opposite forces on $z$ and $z'$ and drives them apart, so that the difference $d$ ceases to be redundant and begins to carry information about the loss landscape.

## D. Proofs for Dyadic EP

We now demonstrate that Dyadic EP correctly trains the parameters $\theta$ of the original force field $F(x, \theta)$, giving the exact gradient $\frac{\mathrm{d}C(\bar{x}^0)}{\mathrm{d}\theta}$ in the limit of infinitesimal nudging.

### D.1. Proof of EP

First, recall that standard EP does not strictly require the system to settle at an energy minimum; it requires only that the system reaches a stationary state (a fixed point of the dynamics). Indeed, using the notation of Section 2.1, EP relies on the key identity:

$$\frac{\mathrm{d}^2}{\mathrm{d}\theta \mathrm{d}\beta} E_T(\bar{x}^\beta, \theta) = \frac{\mathrm{d}^2}{\mathrm{d}\beta \mathrm{d}\theta} E_T(\bar{x}^\beta, \theta). \quad (57)$$

Expanding the total derivative with respect to $\beta$ gives:

$$\frac{\mathrm{d}}{\mathrm{d}\beta} E_T(\bar{x}^\beta, \theta) = \left(\frac{\partial E_T(\bar{x}^\beta, \theta)}{\partial x}\right)^\top \frac{\mathrm{d}\bar{x}^\beta}{\mathrm{d}\beta} + \frac{\partial E_T(\bar{x}^\beta, \theta)}{\partial \beta}$$

$$= C(\bar{x}^\beta). \quad (58)$$

Where the first term vanishes because the system is at a stationary state, *i.e.*, $\frac{\partial}{\partial x} E_T(\bar{x}^\beta, \theta) = 0$; this holds even if the system is not at a minimum of $E_T$. Similarly, for the derivative with respect to $\theta$:

$$\frac{\mathrm{d}}{\mathrm{d}\theta} E_T(\bar{x}^\beta, \theta) = \frac{\partial E_T(\bar{x}^\beta, \theta)}{\partial \theta}, \quad (59)$$

where we additionally assume that the cost function does not depend explicitly on the parameters $\theta$. Substituting these results into Eq. (57) in the limit of infinitesimal nudging ($\beta \to 0$) recovers the fundamental relation given by Eq. (9).

### D.2. Proof of Dyadic EP

We analyze now the stationary states of Dyadic EP by introducing the change of variables:

$$m = \frac{z + z'}{2}, \qquad d = z - z'. \quad (60)$$

In these coordinates, the augmented energy $H_T$ becomes

$$H_T(m, d, \theta, \beta) = -d^\top F(m, \theta) + \beta C(m) \quad (61)$$

and the dynamics in Eq. (28) can be rewritten as:

$$\frac{dm}{dt} = -\frac{\partial H_T}{\partial d} = F(m, \theta), \tag{62}$$

$$\frac{dd}{dt} = -\frac{\partial H_T}{\partial m} = d^T \mathcal{J}_F(m, \theta) - \beta \frac{\partial}{\partial m} C(m). \tag{63}$$

The stationary states $(\overline{m}^\beta, \overline{d}^\beta)$ are the solutions to:

$$F(\overline{m}^\beta, \theta) = 0, \tag{64}$$

$$\overline{d}^{\beta T} \mathcal{J}_F(\overline{m}^\beta, \theta) - \beta \frac{\partial}{\partial m} C(\overline{m}^\beta) = 0. \tag{65}$$

This leads to the following observations:

1) The stationary state of $m$ is independent of $\beta$ and coincides with the stationary state of the original system:

$$\frac{\overline{z}^\beta + \overline{z}'^\beta}{2} = \overline{m}^\beta = \overline{m}^0 = \overline{x}^0 . \tag{66}$$

2) The Jacobian of the extended system defined in Eq. (26) is invertible, provided $\mathcal{J}_F$ is invertible. This is most evident from Eq. (63).

3) The stationary state value of $d$ is given by:

$$\overline{d}^\beta = \beta \left( \mathcal{J}_F^\top(\overline{m}^0, \theta) \right)^{-1} \left( \frac{\partial C}{\partial x}(\overline{x}^0) \right) \tag{67}$$

In particular, when $\beta = 0$, we have $\overline{d}^0 = 0$, which implies that the free stationary states coincide: $\overline{z}^0 = \overline{z}'^0$.

4) The cost at the stationary state of the extended system is equal to the cost at the stationary state of the original system:

$$D(\overline{m}^0) = C(\overline{x}^0). \tag{68}$$

Consequently, the gradients of the cost with respect to the parameters are identical.

Since both the original and extended systems, given respectively in Eq. (28) and Eq. (1-2), share the same cost at their respective stationary states, and because the Jacobians of both models are invertible, applying EP update rule to the extended system give the correct gradient estimate for the parameters $\theta$ of the original system.

The final step of the proof is to establish the equivalence between the standard parameter update rule in Eq. (8) and the modified rule used by Dyadic EP in Eq. (34). Indeed, if we were to apply the standard update rule in the extended space, the update would be:

$$\Delta\theta \propto -\frac{1}{2\beta} \left( \frac{\partial H(\overline{z}^\beta, \overline{z}'^\beta, \theta)}{\partial \theta} - \frac{\partial H(\overline{z}^{-\beta}, \overline{z}'^{-\beta}, \theta)}{\partial \theta} \right). \tag{69}$$

In Dyadic EP, we instead employ the single-phase update:

$$\Delta\theta \propto -\frac{1}{\beta} \left( \frac{\partial H(\overline{z}^\beta, \overline{z}'^\beta, \theta)}{\partial \theta} \right) \tag{70}$$

This choice avoids the overhead of evolving two coupled equations in the extended space, which would be computationally equivalent to evolving four equations in the original space (two for $+\beta$ and two for $-\beta$). Using Eq. (70), we evolve only one coupled equation for $+\beta$ in the extended space; this corresponds to two equations in the original space, thereby achieving the same computational complexity as AsymEP. Furthermore, this single-phase formulation suggests a pathway toward making the update local in time, provided appropriate hardware is used to implement the augmented phase.

Mathematically, these two approaches yield the same gradient estimate because the equations for $d^\beta$ are linear. Explicitly we have :

$$\frac{\partial H(\overline{z}^\beta, \overline{z}'^\beta, \theta)}{\partial \theta} = -(\overline{z}^\beta - \overline{z}'^\beta)^\top \frac{\partial F}{\partial \theta} \left( \frac{\overline{z}^\beta + \overline{z}'^\beta}{2}, \theta \right)$$

$$= -\beta \left( \frac{\partial F}{\partial \theta} \left( \overline{z}^0, \theta \right) \right)^\top \left( \mathcal{J}_F^\top(\overline{z}^0, \theta) \right)^{-1}$$

$$\times \left( \frac{\partial C}{\partial x}(\overline{z}^0) \right), \tag{71}$$

where we have used Eqs. (66) and (67). Inspection of Eq. (71) confirms that, up to corrections of order $\beta^2$, we obtain exactly the same gradient as in AsymEP.

## E. AsymEP versus Dyadic EP

In this appendix, we demonstrate that Asymmetric Equilibrium Propagation (AsymEP) emerges naturally as the first-order projection of the $2N$-dimensional Dyadic Equilibrium Propagation onto a single $N$-dimensional state space. We then formalize the physical trade-offs between the two architectures.

### E.1. AsymEP as the Linear Projection of Dyadic EP

As established in Appendix D.2, transforming the $2N$-dimensional extended space $(z, z')$ into the mean state $m = \frac{z+z'}{2}$ and the difference state $d = z - z'$ exactly decouples the stationary dynamics. Because the stationary state of $m$ is the free state of the original system ($\overline{m}^\beta = \overline{x}^0$), the cost function drives the difference variable to a stationary state $\overline{d}^\beta$ satisfying:

$$\mathcal{J}_F^\top(\overline{x}^0, \theta)\overline{d}^\beta = \beta \frac{\partial C}{\partial x}(\overline{x}^0) \tag{72}$$

To recover this exact error signal in an $N$-dimensional space, we postulate a modified dynamical system $F_A(x)$ compris-

ing the standard EP dynamics and a spatial correction $\Gamma(x)$:

$$F_A(x) = F(x) - \beta \frac{\partial C}{\partial x}(x) + \Gamma(x) \qquad (73)$$

Let $\Delta x = \overline{x}_A^\beta - \overline{x}^0$ denote the displacement from the free equilibrium. Expanding the stationarity condition $F_A(\overline{x}_A^\beta) = 0$ to first order around $\overline{x}^0$ yields:

$$\mathcal{J}_F(\overline{x}^0, \theta)\Delta x - \beta \frac{\partial C}{\partial x}(\overline{x}^0) + \Gamma(\overline{x}_A^\beta) \approx 0 \qquad (74)$$

To ensure the first-order displacement matches the Dyadic EP error signal (i.e., $\Delta x \approx \overline{d}^\beta$), we substitute Eq. (72) into the expansion:

$$\Gamma(\overline{x}_A^\beta) = \left( \mathcal{J}_F^\top(\overline{x}^0, \theta) - \mathcal{J}_F(\overline{x}^0, \theta) \right) \Delta x \qquad (75)$$

$$= -2A_\mathcal{J}(\overline{x}^0, \theta)(\overline{x}_A^\beta - \overline{x}^0) \qquad (76)$$

This uniquely recovers the AsymEP augmented dynamics. Finally, to eliminate the $\mathcal{O}(\beta^2)$ error, AsymEP evaluates the centered difference of two opposite nudges:

$$\overline{x}_A^{\pm\beta} = \overline{x}^0 \pm \beta \left. \frac{d\overline{x}_A}{d\beta} \right|_{\beta=0} + \mathcal{O}(\beta^2) \qquad (77)$$

Subtracting these states cancels the $\mathcal{O}(\beta^2)$ error, yielding $\frac{1}{2}(\overline{x}_A^{+\beta} - \overline{x}_A^{-\beta}) = \overline{d}^\beta + \mathcal{O}(\beta^3)$, successfully recovering the exact post-synaptic update term.

### E.2. Physical Trade-offs and the Extended Space

We can view AsymEP and Dyadic EP as a space-time trade-off of the same underlying physical optimization problem.

AsymEP preserves the original $N$-dimensional state space of the network at the cost of temporal non-locality. The system must evolve sequentially, requiring physical memory not only to store the free equilibrium $\overline{x}^0$ for the asymmetric correction, but also to store the successive stationary states required to evaluate the contrastive gradient update. AsymEP thus serves as the direct, spatially minimal extension of EP.

Dyadic EP provide a learning signal that is local in both space (where $z - z'$ encodes the gradient) and time (allowing the nudged phases to execute in parallel) at the cost of doubling the state space. In particular, capturing non-conservative forces in this extended space requires a specific bilinear coupling, rather than a trivial superposition of uncoupled subsystems. It can be seen as a blueprint for future neuromorphic hardware.

Ultimately, the reduction of Dyadic EP to AsymEP via the variables $m$ and $d$ proves the universality of EP's variational principle.

## F. Derivation of the Hopfield-like Energy

In this section, we derive the explicit energy functional for the Continuous Asymmetric Hopfield dynamics defined in Eq. (35). The force field is given by:

$$F(x) = \rho'(x) \odot (J\rho(x)) - x. \qquad (78)$$

We omit external inputs $J^{\text{in}}$ for brevity, as they appear symmetrically in the Jacobian. The variational Hamiltonian is defined as:

$$H(z, z') = -(z - z')^\top F\left(\frac{z + z'}{2}\right) + \beta C\left(\frac{z + z'}{2}\right). \qquad (79)$$

To analyze this expression, we introduce the midpoint $m = \frac{z+z'}{2}$ and the difference $d = z - z'$. Since the separation between $z$ and $z'$ is induced solely by the nudging parameter $\beta$, the difference scales as $\|d\| \sim \mathcal{O}(\beta)$. We therefore neglect terms of order $\mathcal{O}(\|d\|^3)$ (i.e., or equivalently $\mathcal{O}(\beta^3)$) as they do not contribute to the gradient of the cost.

The activation at the midpoint can be approximated as:

$$\rho(m) = \frac{\rho(z) + \rho(z')}{2} + \mathcal{O}(\|d\|^2). \qquad (80)$$

Similarly, the difference in activations is:

$$\rho(z) - \rho(z') = \rho'(m) \odot d + \mathcal{O}(\|d\|^3). \qquad (81)$$

Inverting this relation, we express the state difference as:

$$z - z' = (\rho(z) - \rho(z')) \odot \rho'(m) + \mathcal{O}(\|d\|^3). \qquad (82)$$

We substitute these expansions into the interaction term of the Hamiltonian, $H_{\text{int}} = -(z - z')^\top (\rho'(m) \odot J\rho(m))$. Applying the identity $a^\top(b \odot c) = (a \odot b)^\top c$, we obtain:

$$H_{\text{int}} = -\left( (z - z') \odot \rho'(m) \right)^\top J\rho(m)$$

$$\approx -(\rho(z) - \rho(z'))^\top J\left(\frac{\rho(z) + \rho(z')}{2}\right). \qquad (83)$$

Expanding the product gives:

$$H_{\text{int}} = -\frac{1}{2}\Big[ \rho(z)^\top J\rho(z) + \rho(z)^\top J\rho(z')$$
$$- \rho(z')^\top J\rho(z) - \rho(z')^\top J\rho(z') \Big]. \qquad (84)$$

We decompose the connectivity matrix $J$ into its symmetric part $S$ and antisymmetric part $A$. The first and last terms simplify to $\rho(z)^\top S\rho(z)$. The cross terms satisfy:

$$\rho(z)^\top J\rho(z') - \rho(z')^\top J\rho(z) = \rho(z)^\top (J - J^\top)\rho(z')$$
$$= \rho(z)^\top (2A)\rho(z'). \qquad (85)$$

Thus, the interaction term reduces to:

$$H_{\text{int}} = -\frac{1}{2}\rho(z)^\top S\rho(z) + \frac{1}{2}\rho(z')^\top S\rho(z')$$
$$- \rho(z)^\top A\rho(z') + \mathcal{O}(\|d\|^3). \tag{86}$$

Finally, for the nudging term, we expand the cost function around the midpoint:

$$C(m) = \frac{1}{2}(C(z) + C(z')) + \mathcal{O}(\|d\|^2). \tag{87}$$

When multiplying by $\beta$, the remainder term becomes $\beta \cdot \mathcal{O}(\|d\|^2)$. Since $\|d\| \sim \mathcal{O}(\beta)$, this remainder is of order $\mathcal{O}(\beta^3)$ and can be consistently discarded alongside the third-order terms from the interaction expansion.

Combining all these components, the final Hamiltonian is:

$$H(z, z') = -\frac{1}{2}\rho(z)^\top S\rho(z) + \frac{1}{2}\rho(z')^\top S\rho(z')$$
$$- \rho(z)^\top A\rho(z') + \frac{1}{2}(\|z\|^2 - \|z'\|^2)$$
$$+ \frac{\beta}{2}(C(z) + C(z')). \tag{88}$$

The saddle-point dynamics, given by Eq. 32, generated by this Hamiltonian are:

$$\frac{\mathrm{d}z}{\mathrm{d}t} = \rho'(z) \odot (S\rho(z) + A\rho(z')) - z - \frac{\beta}{2}\frac{\partial C}{\partial z}, \tag{89}$$

$$\frac{\mathrm{d}z'}{\mathrm{d}t} = \rho'(z') \odot (S\rho(z') + A\rho(z)) - z' + \frac{\beta}{2}\frac{\partial C}{\partial z'}. \tag{90}$$

This system recovers the original continuous Hopfield dynamics when $z = z'$ (assuming $\beta = 0$).

# G. Experimental Details

As in the main text, the neuronal dynamics are governed by the vector field:

$$F_i = \rho'(x_i)\left(\sum_j J_{ij}^{\text{dyn}}\rho(x_j) + b_i(u)\right) - x_i, \tag{91}$$

where the input-dependent bias $b_i(u)$ is precomputed for each MNIST input $u$ as:

$$b_i(u) = \sum_{l\in\text{in}} J_{il}^{\text{in}}u_l. \tag{92}$$

This term projects the input space into the recurrent subspace. The bias yields a diagonal contribution to the Jacobian $\mathcal{J}_F = \frac{\partial F}{\partial x}$, and therefore does not contribute to the antisymmetric correction used in the augmented dynamics Eq. (20) of AsymEP.

The input parameters are then updated using the standard learning rule (21). In particular, the presynaptic term associated with the input weights is given by,

$$\frac{\partial F_i}{\partial J_{kl}^{\text{in}}} = \delta_{ik}\rho'(x_i)u_l. \tag{93}$$

The presynaptic terms associated with the dynamical parameters $J_{ij}^{\text{dyn}}$ depend on the experiment.

## G.1. Symmetric Initialization

### G.1.1. LEARNING RULES

For clarity, we write the learning rules for VF and AsymEP. For the input weights, using (93), we have:

$$\Delta J_{ik}^{\text{in}} \propto \frac{1}{2\beta}\left[(\overline{x}_i^{+\beta} - \overline{x}_i^{-\beta})\rho'(\overline{x}_i^0)u_k\right], \tag{94}$$

while for the recurrent weight, we get:

$$\Delta J_{ij}^{\text{dyn}} \propto \frac{1}{2\beta}\left[(\overline{x}_i^{+\beta} - \overline{x}_i^{-\beta})\rho'(\overline{x}_i^0)\rho(\overline{x}_j^0)\right]. \tag{95}$$

For EP, we have:

$$\Delta J_{ik}^{\text{in}} \propto \frac{1}{2\beta}\left[\left(\rho(\overline{x}_i^{+\beta}) - \rho(\overline{x}_i^{-\beta})\right)u_k\right], \tag{96}$$

and for the recurrent weights:

$$\Delta J_{ij}^{\text{dyn}} \propto \frac{1}{2\beta}\left[\rho(\overline{x}_i^{+\beta})\rho(\overline{x}_j^{+\beta}) - \rho(\overline{x}_i^{-\beta})\rho(\overline{x}_j^{-\beta})\right]. \tag{97}$$

### G.1.2. SUPPLEMENTARY NUMERICAL RESULTS

To complement Fig. 2, we report the evolution of the accuracy of the three methods in Fig. 4. We consider a layered network with 50 hidden neurons. While this capacity is insufficient for state-of-the-art performance, it amplifies the difference in accuracy between models to aid visualization. Models are trained for 20 epochs starting from a symmetric configuration, the natural setting for both VF and EP. With this initialization, AsymEP consistently outperforms the other methods and learns faster by exploiting the additional degrees of freedom of the asymmetric network.

## G.2. Fixed Asymmetry Ratio

This section details the implementation for the fixed asymmetry ratio experiments presented in Section 5.2, followed by complementary numerical results regarding learning speed and induced Jacobian asymmetry.

### G.2.1. LEARNING RULES

**Parametrization and notation.** To enforce a fixed asymmetry ratio, we explicitly parameterize the independent elements of Eq. (38). We introduce two parameter vectors $\theta^S$

| Parameter | Sym. Init. / Feedforward sec. 5.1 & 5.3 | Fixed $r_{\text{str}}$ sec. 5.2 | Fixed $r_{\text{str}}$ & $r_{\text{in}}$ app. G.3 |
| --- | --- | --- | --- |
| Learning Rate (Input-Hidden) | 0.05 | 0.05 | 0.0125 |
| Learning Rate (Hidden-Output) | 0.01 | 0.01 | 0.0025 |
| Time Step (Dynamics Integration) | 0.5 | 0.3 | 0.3 |
| Nudging Parameter ($\beta$) | 0.5 | 0.5 | 0.5 |
| Free-phase Steps ($n_{\text{free}}$) | 20 | 30 | 40 |
| Nudged-phase Steps ($n_{\text{nudge}}$) | 10 | 10 | 10 |
| Number of Epochs | 40 / 20 | 30 | 40 |
| Batch Size | 64 | 64 | 64 |
| Scaling Parameter $\gamma$ | n.a. | $\sqrt{60}$ | $\sqrt{60}$ |
| Structure | 784 - n.a. -10 | 784-50-10 | all-to-all, 500 hid |
| Activation function $\rho$ | tanh | tanh | tanh |
| Initial Recurrent State $s$ | $s \sim \mathcal{U}(-1,1)$ | $s \sim \mathcal{U}(-1,1)$ | $s \sim \mathcal{U}(-1,1)$ |
| Initial Parameters $\theta$ | $\theta \sim \mathcal{N}(0, \frac{1}{N})$ | $\theta \sim \mathcal{N}(0, \frac{1}{N})$ | $\theta \sim \mathcal{N}(0, \frac{1}{N})$ |
| Number of Runs (training + inference) | 10 | 10 | 10 |

*Table 3.* Trained Model Hyperparameters on MNIST. $N$ is the total number of neurons, $\mathcal{U}(-1,1)$ is a uniform distribution, and $\mathcal{N}(\mu, \sigma^2)$ is a Gaussian distribution. For the $r_{\text{str}}$ parametrization, we choose more cautious hyperparameters for training and inference compared to the symmetric initialization, due to increasingly non-conservative and potentially oscillatory dynamics.

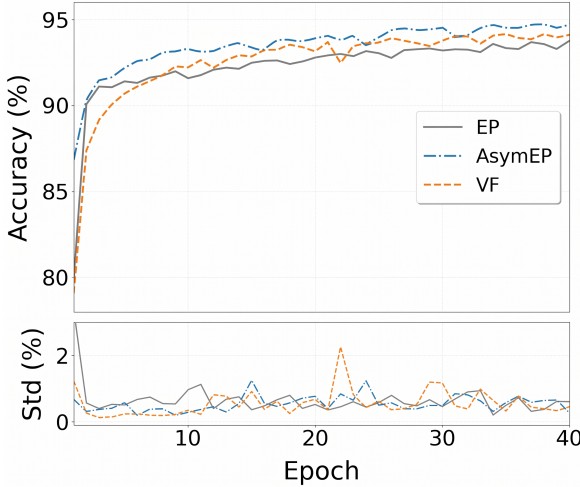

*Figure 4.* Evolution of the mean accuracy and standard deviation (over 10 runs) during training on MNIST for AsymEP, EP, and VF. Models use 50 hidden neurons.

and $\theta^A$ of size $M = N_{\text{dyn}}(N_{\text{dyn}} - 1)/2$, which encode the off-diagonal elements of the symmetric and antisymmetric components $\tilde{S}$ and $\tilde{A}$, respectively. The correspondence between matrix and vector indices is given by:

$$k(i,j) = \frac{(i-1)(i-2)}{2} + j, \quad (1 \le j < i \le N_{\text{dyn}})$$

(98)

where the condition $j < i$ selects the strictly lower triangular elements. Introducing an additional vector $\xi$ for the diagonal elements of $\tilde{S}$, the full matrices are constructed as:

$$\tilde{S}_{ij} = \delta_{ij}\xi_i + (1 - \delta_{ij})\theta^S_{k(\max(i,j),\min(i,j))},$$

(99)

$$\tilde{A}_{ij} = \epsilon_{ij}\theta^A_{k(\max(i,j),\min(i,j))},$$

(100)

where $\epsilon_{ij}$ is the Levi-Civita symbol. The dynamical parameters are then given by:

$$J^{\text{dyn}}_{ij} = \gamma(c_S \tilde{S}_{ij} + c_A \tilde{A}_{ij}),$$

(101)

with normalization coefficients

$$c_S = \frac{\sqrt{1 - r^2_{\text{str}}}}{F_S}, \qquad c_A = \frac{r_{\text{str}}}{F_A},$$

(102)

defined in terms of the Frobenius norms:

$$F_S = \sqrt{\sum_{i=1}^{N} \xi_i^2 + 2\sum_{k=1}^{M} \left(\theta^S_k\right)^2},$$

(103)

$$F_A = \sqrt{2\sum_{k=1}^{M} \left(\theta^A_k\right)^2}.$$

(104)

**Presynaptic computation.** The dependence of the normalization coefficients on the parameters introduces additional regularization terms in the learning rule compared to the parameterization of (Scellier & Bengio, 2017). The gradients of the normalization coefficients are:

$$\frac{\partial c_S}{\partial \theta^S_k} = -2c_S \frac{\theta^S_k}{(F_S)^2}, \qquad \frac{\partial c_S}{\partial \xi_m} = -c_S \frac{\xi_m}{(F_S)^2},$$

(105)

$$\frac{\partial c_A}{\partial \theta^A_k} = -2c_A \frac{\theta^A_k}{(F_A)^2}.$$

(106)

| Parameter | Comparison Dyn. sec. 5.4 | 2 hidden layers sec. 5.4 | 3 hidden layers sec. 5.4 |
|---|---|---|---|
| Learning Rate (Input-Hidden) | 0.0016 | 0.0013 | 0.6 |
| Learning Rate (Hidden-Hidden) | 0.0016 | 0.0013 | 0.6 |
| Learning Rate (Hidden-Output) | 0.0016 | 0.0013 | 0.6 |
| Time Step (Dynamics Integration) | 0.4 | 0.3 | 0.0075 |
| Nudging Parameter ($\beta$) | 0.3 | 0.5 | 0.20 |
| Free-phase Steps ($n_{\text{free}}$) | 40 | 40 | 60 |
| Nudged-phase Steps ($n_{\text{nudge}}$) | 20 | 20 | 30 |
| Number of Epochs | 50 | 40 | 40 |
| Batch Size | 64 | 64 | 64 |
| Layer Structure | 784-500-200-10 | 784-500-500-10 | 784-500-500-500-10 |
| Activation function $\rho$ | tanh | tanh | tanh |
| Initial Recurrent State $s$ | $s \sim \mathcal{U}(-1,1)$ | $s \sim \mathcal{U}(-1,1)$ | $s \sim \mathcal{U}(-1,1)$ |
| Initial Parameters $\theta$ | $\theta \sim \mathcal{N}(0, \frac{1}{N})$ | $\theta \sim \mathcal{N}(0, \frac{1}{N})$ | $\theta \sim \mathcal{N}(0, \frac{1}{N})$ |
| Number of Runs (training + inference) | 10 | 10 | 10 |

*Table 4.* Trained Model Hyperparameters on Fashion-MNIST. $N$ is the total number of neurons, $\mathcal{U}(-1,1)$ is a uniform distribution, and $\mathcal{N}(\mu, \sigma^2)$ is a Gaussian distribution. For the $r_{\text{str}}$ parametrization, we choose more cautious hyperparameters for training and inference compared to the symmetric initialization, due to increasingly non-conservative and potentially oscillatory dynamics.

Combining these with the derivatives of the matrices $\tilde{S}$ and $\tilde{A}$, we have:

$$\frac{\partial \tilde{S}_{ij}}{\partial \theta_k^S} = \delta_{ip}\delta_{jq} + \delta_{iq}\delta_{jp}, \qquad \frac{\partial \tilde{S}_{ij}}{\partial \xi_k} = \delta_{ij}\delta_{kj} \qquad (107)$$

$$\frac{\partial \tilde{A}_{ij}}{\partial \theta_k^A} = \delta_{ip}\delta_{jq} - \delta_{iq}\delta_{jp}, \qquad (108)$$

where $k$ corresponds to the index pair $(p,q)$ with $p > q$, as defined in Eq. (98). The full presynaptic terms are then:

- For the diagonal parameters $\xi_m$:

$$\frac{\partial F_i}{\partial \xi_m} = \gamma c_S \rho'(x_i)\left[-\frac{\xi_m}{(F_S)^2}\sum_{j=1}^N \tilde{S}_{ij}\rho(x_j) + \delta_{im}\rho(x_m)\right]. \qquad (109)$$

- For the off-diagonal symmetric parameters $\theta_k^S$ (where $p > q$):

$$\frac{\partial F_i}{\partial \theta_k^S} = \gamma c_S \rho'(x_i)\left[-2\frac{\theta_k^S}{(F_S)^2}\sum_{j=1}^N \tilde{S}_{ij}\rho(x_j) + \delta_{ip}\rho(x_q) + \delta_{iq}\rho(x_p)\right]. \qquad (110)$$

- For the off-diagonal antisymmetric parameters $\theta_k^A$

(where $p > q$):

$$\frac{\partial F_i}{\partial \theta_k^A} = \gamma c_A \rho'(x_i)\left[-2\frac{\theta_k^A}{(F_A)^2}\sum_{j=1}^N \tilde{A}_{ij}\rho(x_j) + \delta_{ip}\rho(x_q) - \delta_{iq}\rho(x_p)\right]. \qquad (111)$$

**Initialization.** To ensure the stability of the system, we initialize our parameters such that the variance of dynamical parameters scales as $\mathbb{Var}\left[J_{ij}^{\text{dyn}}\right] \propto 1/N_{\text{dyn}}$. This is a conservative choice for the layered architectures used in our experiments, where many entries of $J_{ij}^{\text{dyn}}$ are zero.

In practice, we initialize the parameter vectors $\theta^S, \theta^A$, and $\xi$ with identical variances $\sigma^2$. For large $N_{\text{dyn}}$, the expected Frobenius norms approximate to $\mathbb{E}[F_{S,A}] \approx N_{\text{dyn}}\sigma$. Consequently, the normalization coefficients become:

$$c_S \approx \frac{\sqrt{1-r_{\text{str}}^2}}{N_{\text{dyn}}\sigma}, \qquad c_A \approx \frac{r_{\text{str}}}{N_{\text{dyn}}\sigma}. \qquad (112)$$

Since the symmetric and antisymmetric components are statistically independent, the variance of the weights is derived as follows:

- Diagonal elements ($i = j$):

$$\mathbb{Var}\left[J_{ii}^{\text{dyn}}\right] = \gamma^2 c_S^2 \sigma^2 \approx \gamma^2 \frac{1-r_{\text{str}}^2}{N_{\text{dyn}}^2}. \qquad (113)$$

- Off-diagonal elements ($i \neq j$):

$$\mathbb{V}\mathrm{ar}\left[J_{ij}^{\mathrm{dyn}}\right] = \gamma^2 \left(c_S^2 + c_A^2\right) \sigma^2 \approx \frac{\gamma^2}{N_{\mathrm{dyn}}^2}, \qquad (114)$$

To satisfy $\mathbb{V}\mathrm{ar}\left[J_{ij}^{\mathrm{dyn}}\right] \propto 1/N_{\mathrm{dyn}}$, we set:

$$\gamma = \sqrt{N_{\mathrm{dyn}}} \qquad (115)$$

Note that by random matrix theory, diagonal elements do not affect stability in the large $N_{\mathrm{dyn}}$ limit.

**Potential Simplification.** Although the parameterization above is fully general, a simpler construction is possible by removing self-connections ($\xi = 0$) and enforcing identical parameterization for the symmetric and antisymmetric components, *i.e.*, $\theta^S = \theta^A = \theta$. The matrix elements then become:

$$\tilde{S}_{ij} = (1 - \delta_{ij})\theta_{k(\max(i,j),\min(i,j))}, \qquad (116)$$
$$\tilde{A}_{ij} = \epsilon_{ij}\theta_{k(\max(i,j),\min(i,j))}. \qquad (117)$$

In this case, the Frobenius norms are equal ($F_S = F_A$), and we can omit the explicit normalization:

$$J_{ij}^{\mathrm{dyn}} = \sqrt{1 - r_{\mathrm{str}}^2}\,\tilde{S}_{ij} + r_{\mathrm{str}}\tilde{A}_{ij}. \qquad (118)$$

For a parameter $\theta_k$ corresponding to indices $(p, q)$ with $p > q$, the presynaptic term is given by:

$$\frac{\partial F_i}{\partial \theta_k} = \rho'(x_i)\left[\left(\sqrt{1 - r_{\mathrm{str}}^2} + r_{\mathrm{str}}\right)\delta_{ip}\rho(x_q)\right.$$
$$\left. + \left(\sqrt{1 - r_{\mathrm{str}}^2} - r_{\mathrm{str}}\right)\delta_{iq}\rho(x_p)\right]. \qquad (119)$$

While this parameterization works in simulations and keeps the number of parameters constant for all $r_{\mathrm{str}}$, it constrains the asymmetry to be "homogeneous", by which we mean that the asymmetry ratio is identical for every pair of neurons; hence, the network cannot learn to be symmetric in one region and antisymmetric in another. Therefore, we choose to explore the more general case of (38) in our experiments.

### G.2.2. SUPPLEMENTARY NUMERICAL RESULTS

To complement the results of Fig 3, we analyze the training efficiency as a function of the asymmetry ratio $r_{\mathrm{str}}$ and investigate the robustness of VF by monitoring the Jacobian asymmetry.

**Training efficiency.** We first study the training efficiency of the two algorithms as a function of the asymmetry ratio $r_{\mathrm{str}}$. Inspired by the related concept in (Cesa-Bianchi &Lugosi, 2006), we define the cumulative loss as the accumulated difference between the free equilibrium cost and

a zero-cost baseline (perfect prediction) during learning. Specifically, for each method and value of $r_{\mathrm{str}}$, we calculate the cumulative loss by summing the batch-averaged costs of the first 5 epochs (out of 30, to avoid saturation effects), and reporting the mean and standard deviation over 10 independent training runs. Mathematically, for each run:

$$\text{Cumul. Loss} = \sum_{\text{epoch}=1}^{5} \sum_{k=1}^{N_{\text{batches}}} \left( \sum_{(\overline{x}^0, u) \in \mathcal{B}_k} \frac{C(\overline{x}^0, u)}{|\mathcal{B}_k|} \right), \qquad (120)$$

where $\mathcal{B}_k$ represents the $k$-th batch, and $|\mathcal{B}_k|$ denotes the number of examples in the batch. The parameters are updated after each batch step; consequently, the free equilibrium $\overline{x}^0$ is inferred using the updated parameters and the current example $u$.

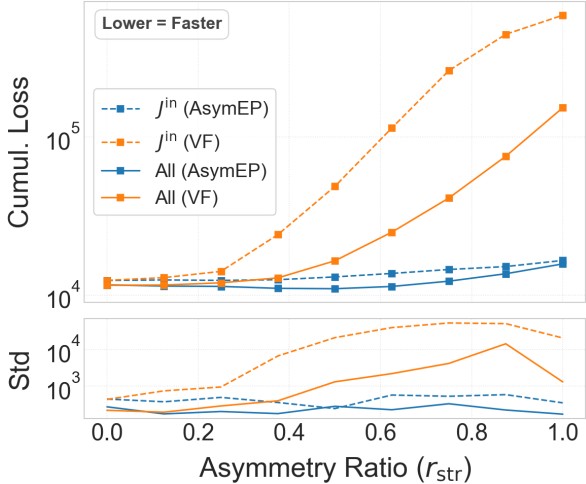

*Figure 5.* Cumulative loss as defined by (120) over the first 5 epochs of training, for different asymmetry ratios $r_{\mathrm{str}}$. We compare VF (orange) and AsymEP (blue), under two training regimes: training only $J^{\mathrm{in}}$ (dashed) or all parameters (solid).

In Fig 5, we observe that learning slows down for both algorithms when $r_{\mathrm{str}} \gtrsim 0.6$. This behavior likely results from the increased difficulty of reaching a stationary state as the dynamics become strongly asymmetric. With a fixed number of inference steps, incomplete convergence degrades the accuracy of the gradient estimates, thereby slowing down the learning. Fig 5 shows that while VF can eventually achieve competitive accuracy, it is consistently slower than AsymEP as soon as asymmetry is introduced.

**Jacobian asymmetry.** We next examine how the structural asymmetry $r_{\mathrm{str}}$ is reflected in the Jacobian of the dy-

namics (35), given by:

$$\frac{\partial F_i}{\partial s_j} = (1 - \delta_{ij})\rho'(x_i)J_{ij}^{\text{dyn}}\rho'(x_j)$$

$$+ \delta_{ij}\left[\rho'(x_i)(J_{ii}^{\text{dyn}}\rho'(x_i)) + \rho''(x_i)b_i - 1\right]. \tag{121}$$

In our layered architecture, the self-connections are zero ($J_{ii}^{\text{dyn}} = 0$). For the following analysis, we neglect all diagonal terms in the Jacobian (including external inputs and potential), since they do not contribute to the antisymmetric correction (20) and thus to the discrepancy between the performance of VF and AsymEP. Consequently, we define the following asymmetry ratio based solely on the off-diagonal Jacobian $\mathcal{J}_{F,\text{off}}$:

$$r_{\text{jac}} = \frac{\|\mathcal{J}_{F,\text{off}} - \mathcal{J}_{F,\text{off}}^{\top}\|_F}{\|\mathcal{J}_{F,\text{off}}\|_F}, \tag{122}$$

The results are presented in Fig 6. For each trained model and ratio $r_{\text{str}}$, we compute $r_{\text{jac}}$ averaged over the stationary states of the first batch (64 images) across 10 independent runs. We observe that when structural asymmetry is strong and all parameters are trained, VF partially compensates for the asymmetry by adjusting the neuronal states. This can be understood by rewriting the ratio as:

$$r_{\text{jac}} = \frac{\left\|\rho'(x_i)\left(J_{ij}^{\text{dyn}} - (J_{ji}^{\text{dyn}})^{\top}\right)\rho'(x_j)\right\|_F}{\left\|\rho'(x_i)J_{ij}^{\text{dyn}}\rho'(x_j)\right\|_F}. \tag{123}$$

Compared to the structural asymmetry ratio in Eq. (37), a value of $r_{\text{jac}} < r_{\text{str}}$ indicates that the neuronal states effectively dampen the structural asymmetry, rendering the dynamics more symmetric. This symmetrization of the Jacobian appears without imposing an additional symmetrization penalty and could be enhanced using the method of (Laborieux &Zenke, 2022). This mechanism likely explains the superior performance of 'All (VF)' compared to '$J^{\text{in}}$ (VF)' in Fig 3, as the former is able to use the additional degrees of freedom to reduce the effective asymmetry at high $r_{\text{str}}$.

### G.3. Stability analysis with Fixed Asymmetry Ratio & Constrained Inputs Projection

A complete stability analysis of the non-conservative dynamics trainable with AsymEP is beyond the scope of this work. Nevertheless, for the class of continuous Hopfield networks considered here, standard arguments from random matrix theory suggest that asymmetry inherently improves asymptotic stability.

In the dynamics defined by Eq. (91), the linear leak term $-x_i$ shifts the spectrum of the system's Jacobian by $-1$.

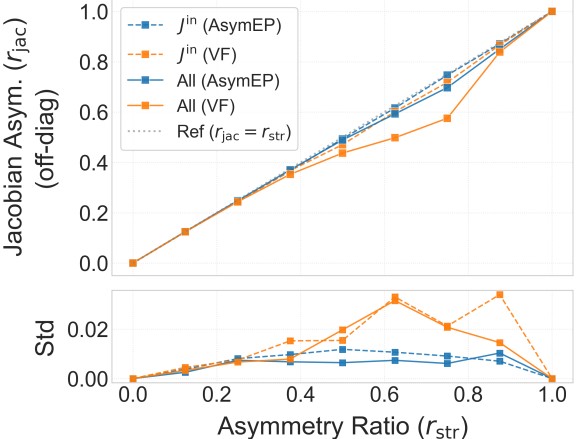

*Figure 6.* Asymmetry ratio of the Jacobian $r_{\text{jac}}$ defined in equation (122) after training for different asymmetry ratios $r_{\text{str}}$. We compare VF (orange) and AsymEP (blue), under two training regimes: training only $J^{\text{in}}$ (dashed) or all parameters (solid).

Consequently, local stability requires the largest real eigenvalue of the effective weight matrix to be strictly less than $1$. Assuming weights are initialized independently with variance $\sigma^2$, Girko's circular law dictates that the eigenvalues of an asymmetric matrix uniformly populate a disk of radius $\sigma\sqrt{n}$ in the complex plane. In contrast, imposing symmetry forces the eigenvalues onto the real line, broadening the spectral radius to $2\sigma\sqrt{n}$ according to Wigner's semicircle law. As a result, asymmetric networks can stably accommodate larger variance in the weight initializations than their symmetric counterparts.

Asymmetry nevertheless introduces imaginary eigenvalues and, consequently, damped oscillations. To study this effect experimentally in a controlled setting, we constrain the input projections $J^{\text{in}}$. In the experiments of the main text, fixing the structural asymmetry ratio $r_{\text{str}}$ still allowed AsymEP to reduce oscillations by aligning and increasing the input projections $J^{\text{in}}$, thereby adding stabilizing diagonal contributions to the Jacobian. To isolate the network's ability to suppress oscillations independently of the magnitude of the input drive, we further constrain the relative scale of $J^{\text{in}}$ and $J^{\text{dyn}}$ by imposing

$$r_{\text{in}} = \frac{\|J^{\text{in}}\|_F}{\|J^{\text{dyn}}\|_F} = \frac{\|J^{\text{in}}\|_F}{\gamma}, \tag{124}$$

where $\|J^{\text{dyn}}\|_F = \gamma$ following Eq. (101). Defining unscaled input projections $\tilde{J}^{\text{in}}$, we set

$$J^{\text{in}} = r_{\text{in}}\gamma\frac{\tilde{J}^{\text{in}}}{\|\tilde{J}^{\text{in}}\|_F} \tag{125}$$

### G.3.1. LEARNING RULES

Reusing the notations of the previous section, we write $J_{il}^{\text{in}} = \gamma c_{\text{in}}\tilde{J}_{il}^{\text{in}}$ with the normalization $c_{\text{in}} = r_{\text{in}}/F_{\text{in}}$, where

$F_{\text{in}} = \|\tilde{J}^{\text{in}}\|_F$. Applying the chain rule yields:

$$\frac{\partial F_i}{\partial \tilde{J}^{\text{in}}_{kl}} = \gamma c_{\text{in}}\rho'(x_i)\left[\delta_{ik}u_l - \frac{\tilde{J}^{\text{in}}_{kl}}{F^2_{\text{in}}}\sum_m \tilde{J}^{\text{in}}_{im}u_m\right]. \quad (126)$$

And for $\gamma$ we have:

$$\frac{\partial F_i}{\partial \gamma} = \frac{1}{\gamma}(F_i + x_i). \quad (127)$$

### G.3.2. SUPPLEMENTARY NUMERICAL RESULTS

Table 5 reports a worst-case control experiment where the structural asymmetry is fixed at $r_{\text{str}} = 0.7$ while the input scale ratio $r_{\text{in}}$ is varied. The experiment uses an all-to-all architecture on MNIST (excluding direct input-to-output connections). The output variance during extended inference (steps 30-50) confirms that the system successfully learns to suppress oscillations even when $r_{\text{in}}$ is severely restricted. Any small residual oscillations can be mitigated by time-averaging over the inference steps.

Finally, $r_{\text{in}}$ can be interpreted as a measure of the external signal magnitude relative to the internal recurrent dynamics. These results indicate that the system remains capable of learning and stabilizing even under a low external input drive. Even when the input projection $\|J^{\text{in}}\|_F$ is 100 times smaller than the recurrent connections $\|J^{\text{dyn}}\|_F$, the network still achieves $36.34 \pm 6.25\%$ accuracy, which is well above chance level ($\sim 10\%$).

### G.4. Feedforward Network

#### G.4.1. LEARNING RULES

For clarity, we write the learning rules for VF and AsymEP in a feedforward architecture with one hidden layer using the notation of Section 5.3. For the input weights connecting to the hidden layer, we get the usual formula:

$$\Delta J^{\text{in}}_{ik} \propto \frac{1}{2\beta}\left[(\overline{h}^{+\beta}_i - \overline{h}^{-\beta}_i)\rho'(\overline{h}^0_i)u_k\right], \quad (128)$$

while for the feedforward weights connecting the hidden to the output layer, we get:

$$\Delta(W_{h\to o})_{ji} \propto \frac{1}{2\beta}\left[(\overline{o}^{+\beta}_j - \overline{o}^{-\beta}_j)\rho'(\overline{o}^0_j)\rho(\overline{h}^0_i)\right]. \quad (129)$$

Note that EP is not applicable in this case.

#### G.4.2. SUPPLEMENTARY NUMERICAL RESULTS

In addition to the final accuracy reported in Sec. 5.3, we show in Fig. 7 the evolution of the accuracy over 20 epochs for AsymEP and VF.

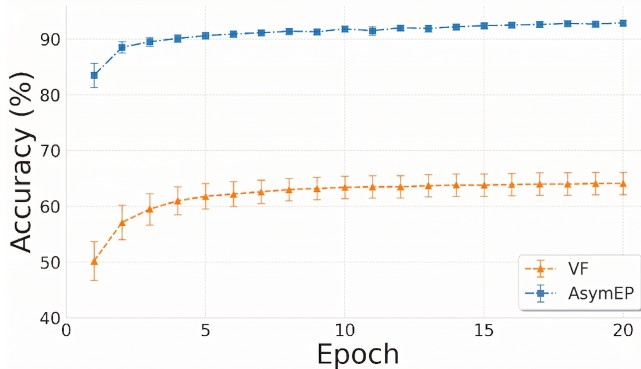

*Figure 7.* Comparison of AsymEP and VF on a feedforward network. Test accuracy on MNIST is shown as a function of training epochs for a single-hidden-layer network with 20 neurons. Curves report the mean and standard deviation over 10 runs. Best accuracies are $92.7\% \pm 0.5\%$ (AsymEP) and $64.3\% \pm 2.0\%$ (VF).

### G.5. Advantages of Non-Conservatives Dynamics

In Section 5.4, we compare three (non-)conservative dynamics under varying constraints. To further evaluate learning speed, Table 6 reports network performance after a single epoch. These results confirm our earlier observation: AsymEP learns faster than VF.

### G.6. Feedforward CIFAR-10 Experiments

This appendix details the architecture and hyperparameters of the deep feedforward experiments comparing backpropagation (BP), VF, AsymEP and Dyadic EP on CIFAR-10 (see subsection 5.5)

**Architecture.** We use a nine-layer convolutional network (denoted CNN9). The first eight layers are convolutional with $3 \times 3$ kernels and zero-padding; spatial downsampling is performed by strided convolutions (stride 2 on layers 2, 4, 6, 8 and stride 1 otherwise), so no pooling is used. The channel widths follow the sequence $3 \to 64 \to 64 \to 128 \to 128 \to 256 \to 256 \to 512 \to 512$, reducing the spatial resolution from $32 \times 32$ to $2 \times 2$. The last layer is a fully connected readout mapping the $512 \times 2 \times 2$ feature map to the 10 class logits. All hidden units use a ReLU nonlinearity. Weights are initialized with the Kaiming scheme ($\sigma = \sqrt{2/\text{fan-in}}$) and biases at zero.

**Training setup.** All methods are trained for 40 epochs with batch size 64 and repeated over 5 seeds. Inputs are normalized per channel and augmented with random $32 \times 32$ crops (padding 4), random horizontal flips and Cutout (one $8 \times 8$ patch). Parameters are updated with SGD with momentum 0.9, weight decay $5 \times 10^{-4}$ and gradient-norm clipping at 1, under a cosine learning-rate schedule decaying from $3.5 \times 10^{-2}$ to $2 \times 10^{-4}$. Test accuracy is reported on an exponential moving average of the weights (decay

*Table 5.* Output variance and final test accuracy on MNIST (%) across different values of $r_{\text{in}}$ with $r_{\text{str}} = 0.7$. (mean $\pm$ std over 10 runs) (500 hiddens, all-to-all, no input-output).

| $r_{\text{in}}$ | Output variance | | Test Acc. (%) |
| | *Untrained* | *Epoch 80* | *Epoch 80* |
| --- | --- | --- | --- |
| 0.01 | $(3.38 \pm 0.90) \times 10^{-4}$ | $(5.22 \pm 2.34) \times 10^{-5}$ | $36.34 \pm 6.25$ |
| 0.10 | $(2.33 \pm 0.48) \times 10^{-4}$ | $(1.39 \pm 0.17) \times 10^{-4}$ | $90.54 \pm 0.19$ |
| 0.50 | $(1.34 \pm 0.32) \times 10^{-5}$ | $(1.06 \pm 0.25) \times 10^{-6}$ | $94.96 \pm 0.10$ |
| 1.00 | $(6.27 \pm 1.24) \times 10^{-7}$ | $(1.75 \pm 0.50) \times 10^{-8}$ | $96.30 \pm 0.09$ |

*Table 6.* Test accuracy on Fashion-MNIST (%) at Epoch 1 (mean $\pm$ std 10 runs). The table compares three classes of network dynamics: Continuous Hopfield (CH), Predictive Coding (PC), and Standard dynamics. Each is evaluated under three connectivity structures: Asymmetric (Asym, $B_k \neq W_{k+1}^{\top}$), Symmetric/conservative (Sym, $B_k = W_{k+1}^{\top}$), and Feedforward (Feedfor, $B_k = 0$).

| | | **EP** | **AsymEP** | **VF** |
| --- | --- | --- | --- | --- |
| | Asym | - | $74.91 \pm 0.45$ | $48.98 \pm 4.09$ |
| CH | Feedfor | - | $74.36 \pm 0.29$ | $48.84 \pm 3.46$ |
| | Sym | $74.57 \pm 0.43$ | - | - |
| PC | Asym | - | $77.83 \pm 0.47$ | $57.75 \pm 3.37$ |
| | Sym | $76.23 \pm 0.39$ | - | - |
| Standard | Asym | - | $76.87 \pm 0.51$ | $61.50 \pm 4.06$ |
| | Feedfor | - | $77.92 \pm 0.51$ | $63.98 \pm 0.73$ |

0.9995). The targets are smoothed ($\varepsilon = 0.1$), which for the EP methods amounts to nudging toward the smoothed one-hot vector.

**Relaxation hyperparameters.** The four methods differ only in the gradient estimator: BP uses automatic differentiation, while the EP-based methods contrast stationary states of the corresponding relaxation dynamics. VF uses an integration step $\eta = 1.0$, nudging $\beta = 0.1$, and at most $K = 1000$ relaxation steps with an early-stopping threshold of $9 \times 10^{-6}$ on the mean state update. Dyadic EP uses the same settings except for a nudging strength $\beta = 0.1$. AsymEP uses a smaller step $\eta = 0.5$, nudging $\beta = 0.1$, and up to $K = 250$ relaxation steps with a threshold of $1 \times 10^{-4}$.

