# OpenReview forum: "Equilibrium Propagation for Non-Conservative Systems"
_ICML.cc/2026/Conference — ICML 2026 regular_

### Official Review · Reviewer_FTzu · 2026-03-04

**Soundness:** 2
**Presentation:** 2
**Significance:** 3
**Originality:** 3
**Overall Recommendation:** 4
**Confidence:** 3

**Summary:**

The authors propose to extend equilibrium propagation (EP) to general dynamical systems of the form $\dot{x}=F(x,\theta,u)$. This paper explores the concept of relaxing the conservative-dynamics assumption required in classical EP, where $F(x)=-\nabla E(x)$ implies symmetric interactions and limits applicability to networks with symmetric weights. To address this limitation, the authors propose two methods: Asymmetric Equilibrium Propagation (AEP), which modifies the nudged dynamics using the antisymmetric part of the Jacobian, and Dyadic EP, which doubles the state variables to construct an energy function whose saddle-point dynamics reproduce the correct gradient behavior. Experimental results on MNIST suggest that the proposed methods outperform the previously proposed vector-field algorithm.

**Compliance With Llm Reviewing Policy:**

Affirmed.

**Final Justification:**

My comments are well addressed and I have updated the score as "4: Weak accept".

**Key Questions For Authors:**

See Weaknesses.

**Limitations:**

Yes

**Strengths And Weaknesses:**

Strengths
- The paper addresses an important limitation of equilibrium propagation (EP). Extending EP beyond conservative dynamics is a meaningful direction, especially given the motivation of enabling learning algorithms that may be compatible with physical or neuromorphic implementations. Developing approaches that could make EP applicable to more general neural network models is therefore a valuable research goal.
- The authors propose two approaches to remove the conservative dynamics assumption in EP. These methods provide concrete ways to extend EP to non-conservative systems, which has been a long-standing limitation of the framework. The proposed constructions appear to be new and conceptually interesting.
- Although the experiments are conducted on relatively small-scale settings, the results demonstrate that the proposed methods outperform the previously proposed vector-field learning approach. This empirical evidence supports the theoretical claims and shows that the methods can work in practice at least in simple scenarios.

Weaknesses
- The paper proposes two approaches, AEP and Dyadic EP, but the relationship between them is not fully clarified. It would help the reader if the authors explicitly explained whether the two methods should be viewed as alternative algorithms with different advantages, or whether Dyadic EP is intended as the main practical method while AEP mainly serves as a conceptual tool for understanding the correction needed in non-conservative systems.
- AEP relies on information derived from the Jacobian of the dynamics. While the paper acknowledges this requirement, it remains unclear how this quantity could be obtained efficiently in practice, especially for large neural networks. It would be helpful for the authors to discuss whether this computation can be approximated locally or whether it introduces significant overhead when scaling to larger models.
Both proposed methods rely on dynamical systems that must relax to steady states. However, the paper does not provide a discussion of the stability or convergence properties of the proposed dynamics. For example, it is unclear under what conditions the dynamics converge to a stable equilibrium and whether oscillatory or unstable behaviors may arise, particularly for non-conservative systems.
- Since the motivation of the work is partly the possibility of physical or neuromorphic implementations, it would strengthen the paper to discuss what would actually be required to realize the proposed algorithms in physical systems. For example, it would be helpful to clarify: how sensitive the algorithm is to noise or imperfect measurements, and what additional assumptions would be needed for a realistic implementation.

---

> ### Author Rebuttal · Authors · 2026-03-31
>
> ## Conceptual Relationship and Implementations: AEP vs. Dyadic EP
>
> We thank the reviewer for this critique. We have updated the manuscript to better illustrate the relationship between AEP and DEP, clarifying that they are alternative algorithms with distinct theoretical and practical merits.
>
> While formally equivalent (both yield the exact same gradient as $\beta \rightarrow 0$), AEP applies a linear correction to the base dynamics, whereas DEP relies on a non-linear correction through the extended space.
>
> * **AEP** operates within the traditional Equilibrium Propagation (EP) framework but extends it to non-conservative systems via a local, linearized correction term in the dynamics. While it preserves core EP advantages, it requires modulating all neurons' internal dynamics during the nudged phase. We view it as a robust extension for existing asymmetric networks. The physical realization of AEP's correction terms is the subject of ongoing work, which we discuss in response to Reviewer Q3zH.
> * **DEP** departs from EP by doubling the state space to provide a learning signal that is local in both space (where $z - z'$ encodes the gradient) and time (allowing the nudged phases to execute in parallel). DEP proves the universality of EP's variational principle, demonstrating that exact gradients for non-conservative systems can be obtained entirely through simultaneous forward-in-time relaxation. It serves as a theoretical blueprint for future neuromorphic hardware.
>
>
> ## AEP Scalability, Stability, Noise
>
> We thank the reviewer for raising these important practical points.
>
> **Scalability.** The complexity of the AEP correction depends on the specific force field. For standard architectures defined by a weight matrix $W$, the Jacobian correction relies entirely on the antisymmetric part, $\frac{W-W^T}{2}$. This computation is spatially local (requiring only $W_{ij}$ and $W_{ji}$) and scales as $O(n^2)$ for a dense layer of $n$ neurons. Since the standard dynamics also require $O(n^2)$ matrix-vector products, AEP introduces no new asymptotic bottleneck.
>
>
> **Stability and Convergence.**
> While hardware stability is beyond our scope, theory suggests asymmetry actually improves asymptotic stability in our continuous Hopfield model; we encountered no convergence issues.
> Theoretically, convergence is guaranteed by the linear leak term ($- x_i$), shifting the Jacobian's eigenvalues into the left half-plane (Gershgorin circle theorem). For large random matrices (of size $n \times n$) with variance $\sigma^2$, the Circular Law bounds the spectral radius to $\sigma\sqrt{n}$. Conversely, forcing symmetry spreads eigenvalues to $2\sigma\sqrt{n}$ (Wigner's Semicircle Law). Consequently, symmetric EP networks require a strictly smaller $\sigma$ for stability than asymmetric ones. However, asymmetry does introduce imaginary eigenvalues and damped oscillations.
>
> Experimentally, AEP mitigates these oscillations by scaling up and learning the input projection ($J^{in}$), adding stabilizing diagonal Jacobian contributions. A worst-case control experiment reported in Table 1
> (Appendix)
> fixes the antisymmetric ratio $r_{str} = 0.7$ and defines input scale $r_{in} = \|{J}^{in}\|_F/\|{J}^{dyn}\|_F$. Output variance during extended inference (steps 30-50) confirms the system learns to suppress oscillations even with severely restricted $r\_{in}$.
> (Table 1).
> Finally, any residual small oscillations can be filtered by time-averaging over inference steps.
>
> **Table 1:** Output variance and final test accuracy on MNIST (\%) across different values of $r_{in}$ with $r_{str} = 0.7$. (mean $\pm$ std over 10 runs).
>
> | **$r_{in}$** | **Output variance** (*Untrained*) | **Output variance** (*Epoch 30*) | **Test Acc. (%)** (*Epoch 30*) |
> | :--- | :--- | :--- | :--- |
> | 0.01 | $(2.17 \pm 3.05) \times 10^{-3}$ | $(5.35 \pm 5.75) \times 10^{-6}$ | $36.43 \pm 7.76$ |
> | 0.10 | $(3.69 \pm 5.59) \times 10^{-3}$ | $(7.24 \pm 1.10) \times 10^{-5}$ | $89.04 \pm 0.23$ |
> | 0.50 | $(6.09 \pm 11.19) \times 10^{-4}$ | $(1.45 \pm 0.35) \times 10^{-5}$ | $92.95 \pm 0.20$ |
> | 1.00 | $(1.32 \pm 1.06) \times 10^{-5}$ | $(3.70 \pm 1.77) \times 10^{-6}$ | $94.25 \pm 0.22$ |
>
> **Noise.** Precise noise modeling is hardware-dependent and left for future work, but several factors suggest robustness. First, EP has been extended to stochastic systems. Second, since VF minimizes cost despite systematic gradient errors in systems with moderate asymmetry, AEP should tolerate zero-mean random noise on corrections. Third, our $r_{in}$ control experiment (Table 1) shows AEP learns even when the input signal is severely attenuated relative to recurrent dynamics.

---

> > ### Author Rebuttal · Reviewer_FTzu · 2026-04-02
> >
> > Thanks for the responses. My comments are well addressed and I have updated the score as "4: Weak accept".

---

### Official Review · Reviewer_USLV · 2026-03-07

**Soundness:** 1
**Presentation:** 3
**Significance:** 2
**Originality:** 4
**Overall Recommendation:** 4
**Confidence:** 3

**Summary:**

Traditional machine learning, particularly deep learning, relies heavily on the **backpropagation (BP)** algorithm for training. However, directly implementing backpropagation in physical systems or neuromorphic hardware presents significant challenges. This is mainly because BP requires **precise weight symmetry** (i.e., the transpose of the weight matrix) and a clearly separated **backward pass**, both of which are difficult to realize in physical systems.

**Equilibrium Propagation (EP)** has been proposed as a promising alternative. EP leverages the **relaxation dynamics of physical systems** to update parameters, avoiding the need for explicit backpropagation. Instead, it estimates gradients by comparing two **perturbed phases** (positive and negative perturbations), enabling efficient approximate learning in **energy-based models**.

**Compliance With Llm Reviewing Policy:**

Affirmed.

**Final Justification:**

The author's experiment has dispelled my concerns.

**Key Questions For Authors:**

One possible additional experiment concerns **training feedforward neural networks with asymmetric dynamics**.

For example, a standard feedforward layer can be written as

$$
x_l = W_l f(x_{l-1})
$$

which can be interpreted as the steady state of the **non-conservative dynamics**

$$
\frac{dx_l}{dt} = -x_l + W_l f(x_{l-1})
$$

In contrast, a symmetric energy-based formulation would correspond to dynamics of the form

$$
\frac{dx_l}{dt} = -\frac{\partial}{\partial x_l}(-x_l + W_l f(x_{l-1}))^2
$$

It would be very interesting to evaluate how different learning methods— **standard backpropagation (BP), AEP, and EP **—perform when training networks defined by these dynamics, respectively.

Such an experiment could provide clearer insight into the practical advantages of the proposed framework. If the authors were able to include such results, it would significantly strengthen the empirical support for the paper. I will increase my score from 3 to 5.

And there are some methods mentioned in [1] can also be used for comparison,  such as PREDICTIVE CODING and CONTRASTIVE HEBBIAN LEARNING.

[1] BACKPROPAGATION AT THE INFINITESIMAL INFER- ENCE LIMIT OF ENERGY-BASED MODELS: UNIFYING PREDICTIVE CODING, EQUILIBRIUM PROPAGATION, AND CONTRASTIVE HEBBIAN LEARNING

**Limitations:**

see weakness

**Strengths And Weaknesses:**

## Strengths

### Core contribution: Two equivalent non-conservative EP algorithms

To address the above limitations, the paper proposes two novel algorithms:
**Asymmetric Equilibrium Propagation (AEP)** and **Dyadic Equilibrium Propagation (Dyadic EP)**.

Both methods are capable of **exactly computing the gradient of the cost function in non-conservative systems** in the limit of vanishing perturbation. This significantly extends the applicability of equilibrium-based learning and provides a principled way to implement learning in a broader class of **physical dynamical systems**.

### Asymmetric Equilibrium Propagation (AEP)

The key idea of **AEP** is to introduce a **local correction term** to the original non-conservative dynamical system. This correction modifies the perturbed dynamics such that the resulting displacement carries the correct error signal.

More specifically, the correction term alters the **Jacobian matrix** of the perturbed system so that, at the free fixed point, it becomes equal to the **transpose of the original Jacobian**. As a result, the displacement
[
(\bar{x}_A^\beta - \bar{x}_A^{-\beta})
]
obtained from the perturbed phases contains the correct **post-synaptic error signal**, ensuring that the resulting update rule approximates the true gradient.

Importantly, the correction term is **spatially local**. This is because the antisymmetric matrix (A) vanishes between neurons that are not directly connected, while the displacement ((x - \bar{x}^0)) is locally available at the synaptic level.

The design of the corresponding energy function is also elegant. Its **saddle-point dynamics**

[
\dot{z} = -\partial_z H_T, \quad \dot{z'} = +\partial_{z'} H_T
]

generate the desired evolution of the system.

---

## Weaknesses

While the proposed framework is conceptually interesting and potentially important, the **empirical evaluation is currently limited**.

The experiments only include a comparison between **AEP and vector field (VF) learning** on the **MNIST handwritten digit classification task** using a continuous asymmetric Hopfield network. This experimental setup is relatively simple and does not sufficiently demonstrate the advantages or broader applicability of the proposed framework.

In my opinion, the paper would benefit from **substantially more experimental validation**. For instance, additional experiments on different architectures or tasks could better illustrate the practical benefits of the method. Expanding the experimental section by roughly a factor of **two to three** would significantly strengthen the paper.

If additional experiments were provided, I would be inclined to increase my score.

---

> ### Author Rebuttal · Authors · 2026-03-31
>
> ### Empirical Evaluation: Asymmetric Feedforward Networks
>
> We thank the reviewer for highlighting the need for broader empirical validation. Alongside new CIFAR-10 and Fashion-MNIST evaluations (see Reviewer Q3zH), we expanded Section 5.3 to compare seven distinct dynamics on Fashion-MNIST.
>
> We use a network with two hidden layers ($x_1, x_2$ sizes 500, 200), input $x_0$, and output $x_3 (= x_L)$. Forward and backward weights are $W_k$ (where $W_1 = J^{in}$, Eq. 35) and $B_k$, respectively. Following the reviewer's suggestion and (Millidge et al, 2022), we evaluate three dynamics:
>
> * **Continuous Hopfield (CH)** ($k > 0$):
>     $$\frac{dx_k}{dt} = -x_k + \rho'(x_k) \odot \left( W_k \rho(x_{k-1}) + (1 - \delta_{k,L}) B_k \rho(x_{k+1}) \right)$$
> * **Predictive Coding (PC)** ($k > 0$) with $e_k = x_k - W_k \rho(x_{k-1})$:
>     $$\frac{dx_k}{dt} = -e_k + (1 - \delta_{k,L}) \left( \rho'(x_k) \odot (B_k e_{k+1}) \right)$$
> * **Standard** ($k > 0$):
>     $$\frac{dx_k}{dt} = -x_k + W_k \rho(x_{k-1}) + (1 - \delta_{k,L}) B_k \rho(x_{k+1})$$
>
> For each dynamics, we examine three connectivity scenarios:
>
> * **Asymmetric**: $B_k \neq W_{k+1}^\top$. To ensure a fair comparison with an identical number of tunable parameters across configurations, we randomly initialize $B_k$ and only train the forward weights $W_k$. Notably for PC, the learning rule for $B_k$ is proportional to $(\bar{e}^0_{k+1})^\top$, which is zero when only the inputs are clamped.
> * **Symmetric (Conservative)**: $B_k = W_{k+1}^\top$. For the CH and PC dynamics, this condition corresponds to the energy functionals: $E_{CH} = \frac{1}{2} \sum_{k=1}^L \|x_k\|^2 - \sum_{k=1}^{L-1} \rho(x_{k+1})^\top W_{k+1} \rho(x_k)$ and $E_{PC} = \frac{1}{2} \sum_{k=1}^L \|e_k\|^2$. For the Standard dynamics, imposing $B_k = W_{k+1}^\top$ still yields a non-symmetric Jacobian, thereby precluding the use of EP.
> * **Feedforward**: $B_k = 0$. Under this condition, the PC and Standard dynamics become identical. As noted in our manuscript, Vector Field (VF) can only train the final layer for this architecture. Note that the learning rule for the Standard dynamics reduces to a form analogous to BP: $\Delta W_k \propto \Delta x_k^{\beta} \, \rho(\bar{x}_{k-1}^0)^\top$. In AEP, the term $\Delta x_k^{\beta} = \frac{1}{2\beta} (x^\beta - x^{-\beta})$ acts as the error term backpropagated through the network from the output in standard BP.
>
> Full learning rules are in the updated appendix. EP, VF, and AEP share identical hyperparameters.
>
> Addressing the request for a BP baseline, evaluating BP on standard feedforward (Cross-Entropy loss, Adam) yields $89.06 \pm 0.33 \%$.
>
> Table 1 shows AEP systematically outperforms VF on asymmetric and feedforward architectures in accuracy and speed. Furthermore, AEP on asymmetric networks surpasses EP on symmetric ones, even when training only forward weights, suggesting an expressivity advantage. Rigorous comparisons with BPTT or RBP are left for future work. AEP also learns faster than VF (averaging a 15% advantage after 1 epoch) and is much more stable (10x variance reduction after 1 epoch).
>
> While focusing on matrix-based networks, our framework is general; it only requires a dynamical system, $\dot{x}=F(x)$, converging to a fixed point. It readily extends to domains like non-equilibrium chemical networks. Practically, dynamics choices depend on hardware, biological plausibility, or computational needs. For instance, the Standard feedforward dynamics allows rapid, one-step free-phase calculation, bypassing the slow post-training inference convergence of CH and PC.
>
> **Table 1: Test accuracy on Fashion-MNIST (%) at Epoch 50 (mean $\pm$ std 10 runs). BP on the standard feedforward using cross entropy and Adam achieve $89.06 \pm 0.33 \%$.**
>
> | Inference | Topology | EP | AEP | VF |
> | :--- | :--- | :--- | :--- | :--- |
> | **CH** | Asymmetric | - | 86.78 $\pm$ 0.14 | 85.20 $\pm$ 0.12 |
> | | Feedforward | - | 86.05 $\pm$ 0.12 | 77.76 $\pm$ 0.37 |
> | | Symmetric | 84.30 $\pm$ 0.13 | - | - |
> | **PC** | Asymmetric | - | 86.20 $\pm$ 0.17 | 80.71 $\pm$ 6.17 |
> | | Symmetric | 84.78 $\pm$ 0.14 | - | - |
> | **Standard** | Asymmetric | - | 82.91 $\pm$ 0.48 | 75.52 $\pm$ 1.69 |
> | | Feedforward | - | 86.25 $\pm$ 0.16 | 78.58 $\pm$ 0.28 |

---

> > ### Author Rebuttal · Reviewer_USLV · 2026-04-03
> >
> > Thank you for the author's reply. The author's experiment has dispelled my concerns. However, could the author explain why BP performs better than the other methods? Is there any intuitive understanding or experimental insight? I suspect that it may be because BP uses Adam, while EP, AEP, and VF all employ similar simple first-order gradient descent. Anyway, EP, AEP, and VF all calculate gradients. After calculating the gradients, can we also use Adam or SGD with momentum to see if the performance will improve for EP, AEP, and VF.

---

> > > ### Author Response · Authors · 2026-04-06
> > >
> > > We thank the reviewer for the remark and continued engagement.
> > >
> > > As you rightly suspected, the initial performance gap was largely driven by the use of Cross-Entropy and Adam in the BP baseline.  To provide a strict apples-to-apples comparison, we have now tuned a BP baseline using MSE and standard SGD (learning rate 0.05). This yields a test accuracy of $87.37 \pm 0.29\%$ (v.s. $86.25
> > > \pm 0.16\%$ for AEP). We will update the manuscript with this baseline.
> > >
> > > The remaining $\sim 1\%$ difference likely is originated by the finite value of $\beta$ (the algorithm gives the true gradient only in the limit $\beta \to 0$) and suboptimal convergence to equilibrium (e.g., due to the Euler discretization scheme) and it is consistent with our new CIFAR-10 results on a deep convolutional network previously illustrated to reviewer Q3zH.

---

### Official Review · Reviewer_KfVN · 2026-03-10

**Soundness:** 3
**Presentation:** 4
**Significance:** 3
**Originality:** 3
**Overall Recommendation:** 5
**Confidence:** 4

**Summary:**

This work tackles the problem of credit assignment in asymmetrically connected fixed point models. The paper proposes two new algorithms related to equilibrium propagation (EP).
The first algorithm behaves as the vector-field (VF) variant of EP during the free phase, but during the nudged phase it applies a new correction term, which is not present in VF. The second variant defines an extended state-space in which positively and negatively perturbed states are equilibrating simultaneously. This allows them to define a scalar potential for networks with asymmetric connectivity. Unlike traditional energy-based models typically explored in EP inference is no longer a pure minimization problem, but a minmax problem.

A drawback for this manuscript is that one of the proposed methods, namely dyadic equilibrium propagation (DEP), already exists in the literature (since 2024 MLNCP) under the very similar name “dyadic learning” (details below). I am not convinced of the soundness of AEP, so I hope the authors will clarify some questions regarding it.

**Compliance With Llm Reviewing Policy:**

Affirmed.

**Final Justification:**

I had a number of concerns regarding soundness (of AEP) and originality (of DEP), which have all been addressed well.

I think this work offers novel theoretical insights. DEP bridges the gap between equilibrium propagation and dyadic learning. This allows you to extend the strong theoretical guarantees of EP to DL and beyond (e.g. to networks with continuous state spaces). AEP introduces a correction term to the nudged phase dynamic of VF style EP, which leads to correct gradient estimates. While in my opinion less elegant (as the dynamics do not cleanly derive from a single scalar potential) the AEP model might map to less complicated hardware (assuming the correction is easy to compute) as you can infer z and z' sequentially on the same physical hardware, whereas DEP seems to require state doubling and some weight sharing between the z and z' components of each dyad.

The main drawback of this manuscript is the limited experimental evaluation, which for asymmetric networks is limited to relatively shallow architectures (the feedforward CNN experiments are deep but bypass the challenges of oscillations as there is no feedback), but I think the theoretical contributions make up for this.

**Key Questions For Authors:**

# Questions regarding AEP
1. The error term you derive in appendix A uses the Neumann expansion and assumes that the asymmetric part of the Jacobian is small.
It seems problematic that the estimate of the error relies on the assumption that the Jacobian is mostly symmetric, when we are trying to quantify the error in the case of general asymmetric Jacobians. In the case where the anti-symmetric component dominates this series will likely diverge, and in the case of a fully anti-symmetric Jacobian it will be undefined wont it? Please let me know if I am misunderstanding something here. It seems like the correction term is not really justified in highly asymmetric setting.

2. One thing that looks strange to me is that for AEP it appears that the nudged fixed point does not coincide with the free fixed point in the limit beta-->0. This is different from DEP, for which the nudged fixed point does converge to the free fixed point in the limit of beta-->0. It seems like the only way to make AEP’s nudged dynamics converge to the free fixed point in the limit beta-->0 is by enforcing that the Jacobian is symmetric. Is this connected to how the “correction term” was derived using an assumption that the skew-symmetric component is small?

# Questions regarding DEP
3. Dyadic equilibrium propagation (DEP) appears to be nearly identical to Dyadic Learning (DL) (Høier et al 2024 MLNCP), though you arrive at the model from a different path. The only difference I can see is that in DL the neuron’s “self-interactions” (the diagonal elements of the Jacobian) are modelled via strictly convex functions G(s)=int_0^s (v)dv), which allows them to use mirror descent updates that avoid the presence of derivatives, whereas you model self-interactions via the squared L2 norm terms on the pre-activations in your objective and perform gradient descent in the pre-activations. Are there significant differences between dyadic learning and dyadic EP? If so then I think that should be discussed. If there is not, then it would be appropriate to use the original (very similar) name and reference the original work.

4. The experiments
- The experiments are very small scale (a single hidden layer with between 20 to 500 units). Did you find it challenging to scale the simulations. Did you have to do anything to avoid limit cycles in the asymmetric models?

### Revising score
If AEP is indeed sound (Q1 and Q2) and if the relationship between dyadic-EP and dyadic learning (Q3) is clearly laid out, then I will adjust my recommendation.

**Limitations:**

The Impact statement is missing. I think it will be sufficient to just add the default impact statement provided by ICML.

**Strengths And Weaknesses:**

# Strengths
- A well written paper with a nice flow.
- Identifies an interesting problem - local learning in asymmetric network - which could have relevance to neuromorphic computing and probably other communities as well.
- Proposes two solutions to the problem (though one is already known and I suspect there might be an issue with the other one).
- Establishes a new connection between the dyadic model and equilibrium propagation.

# Weaknesses
- The derivation of the AEP algorithm seems to hinge on the assumption that the anti-symmetric component of the Jacobian is small (see question below).
- The dyadic EP formulation is nearly identical to dyadic learning ([Høier et al 2024 MLNCP](https://openreview.net/forum?id=LNfWowAErI)) . The only difference appears to be how self interactions are modeled and whether gradient descent (this work) or mirror descent is used during inference.
- The experimental evaluation only explores AEP (If I am mistaken and AEP is sound and equivalent to DEP, then this is not actually a weakness (see questions below)).
- The experiments are very small scale, with the vast majority of learnable parameters in the input projection. This might hide some of the issues you often find in asymmetric Hopfield networks (limit cycles and instability).

# Other comments:
## Regarding related works
- You have misunderstood the cited work dualprop as being energy based and symmetrically connected. It actually corresponds to dyadic learning with a lower triangular (feedforward) Jacobian, for which inference is a minmax problem (so not energy based in the traditional sense). This might be hard to extract from the 2023 dualprop paper, which is quite dense, but it is quite clear from the 2024 dualprop ICML paper and the 2024 MLNCP paper.
## The name AEP
- In the EP literature [Agnostic equilibrium propagation](https://arxiv.org/abs/2205.15021) is also referred to as AEP (and less frequently as Æqprop). To avoid confusion it might be a good idea to choose a different shorthand.
## Sidenote
- This is probably redundant to state, but for the record, I do not see any indication of plagiarism here whatsoever. It is very common in research that the same thing gets discovered independently in multiple labs. The fact that the names dyadic EP and dyadic learning are so similar is not strange either considering that the term “dyadic neuron”  originates in the dualprop paper, which is cited by both this work and the dyadic learning paper.

---

> ### Author Rebuttal · Authors · 2026-03-31
>
> ### Terminology
>
> Regarding potential confusion with Agnostic EP, the original paper used the acronym Æqprop, but we understand the concern and will replace the shorthand with AsymEP in the final version.
>
> ### AEP Soundness: Neumann Expansion & Anti-symmetric Jacobian
>
> There is a crucial distinction we must clarify: the Neumann expansion in Appendix A is **only** used to quantify the gradient estimation error of the baseline *Vector Field (VF)* near the conservative limit. It is **not** used to justify our proposed Asymmetric EP (AEP) algorithm.
>
> The reviewer is correct to note that the Neumann series in Appendix A only converges when the spectral radius $\rho(S_{\mathcal{J}}^{-1}A_{\mathcal{J}}) < 1$ (as hinted in the text), including the limit case of an antisymmetric Jacobian ($S_{\mathcal{J}} = 0$).
> In this case, VF's limitations can be shown without the Neumann expansion: because $J_F = A_J$ and $J_F^\top = -A_J$, VF computes the *negative* true gradient, maximizing the cost. Conversely, AEP computes the exact correct gradient.
>
> ### AEP Soundness: Nudged Fixed Point ($\beta \to 0$)
>
> AEP's nudged fixed point exactly coincides with the free fixed point as $\beta \to 0$. Eq. 20 ensures this:
>
> $$
> F({\overline{x}}_{A}^{\beta}, \theta) - \beta \frac{\partial C}{\partial x}({\overline{x}}_{A}^{\beta}) - 2 A_{\mathcal{J}}({\overline{x}}^{0}, \theta) ({\overline{x}}_{A}^{\beta} - {\overline{x}}^{0}) = 0
> $$
>
> Taking $\beta \to 0$, the equation is solved by the free fixed point $\overline{x}^0$ because $F(\overline{x}^0, \theta) = 0$ and the correction term vanishes: $2A_{\mathcal{J}}(\overline{x}^0, \theta)(\overline{x}^0 - \overline{x}^0) = 0$.
> (notice that in EP the nudged dynamics is initialized with $\overline{x}^0$).
>
> ### Comparison with DL
>
> We thank the reviewer for emphasizing the parallels between Dyadic Equilibrium Propagation (DEP) and Dyadic Learning (DL)(Hoier et al. 2024). Both approaches share a core mechanism of doubling the number of variables. However, DEP emerges from a fundamentally different starting point (as recognized by the reviewer) and provides a more immediate generalization to arbitrary force fields grounded in physical first principles. On the opposite DL proposes a practical and efficient (using mirror descent) implementation for Hopfield networks (Hoier et al. 2024) (asymmetric, feedforward, and very recently convolutional (Nest&Hoier, 2026)). Specifically, the two primary differences are:
>
> First, while DL relies on *a priori* knowledge of local scalar energies for each neuron, DEP starts directly from dynamical systems (arbitrary vector fields) and requires no such assumption. DEP therefore provides a direct, practical, formulation-agnostic strategy to construct a global objective for any force. Additionally, because it does not assume a discrete number of variables, DEP naturally extends to continuous state spaces (e.g., fields rather than particles).
>
> Second, we emphasize the physical origin of our method. Applying the Bateman-Keldysh-Galley formalism to gradient flows (the overdamped limit of classical dynamics) naturally yields our governing equations. Pure gradient flows cannot intrinsically possess a rotational component, yet asymmetric dynamics requires one. By introducing an auxiliary variable to "absorb" this rotation, DEP provides a rigorous physical justification for the extended state space. Adapting this specific framework from classical Lagrangian mechanics to non-conservative gradient flows represents in itself a novel theoretical extension, as confirmed by correspondence with researchers in out-of-equilibrium mechanics.
>
> Finally, we respectfully clarify that DEP is entirely agnostic to the self-interaction term. The $L_2$ norm and gradient descent updates in our current equations simply arise from applying DEP to a Continuous Asymmetric Hopfield network with a linear leak (a natural example for the EP community). Given its generality, DEP could readily be applied to the strictly convex self-interactions proposed in DL.
>
> In the revised version of the manuscript, we will clarify the connection to the works of Høier et al. on Dyadic Learning, citing all the relevant prior works and discussing the differences with our approach. To emphasize this shared extended space ('Dyadic') and our distinct definition of a global energy ('Equilibrium Propagation'), we choose to retain the name DEP.
>
> ### Experimental Stability & Limit Cycles
>
> Expanded Experiments on CIFAR-10 and Fashion-MNIST: see Reviewer Q3zH.
>
> Concerning stability, we added numerical experiments and discussion (see Reviewer FTzu). We neither expect nor observe limit cycles. Empirically, AEP reduces damped oscillations during training without adding Euler discretization instability compared to standard EP.

---

> > ### Author Rebuttal · Reviewer_KfVN · 2026-04-02
> >
> > 1. Thanks for clarifying the connection to DL. I think you are correct. Even though the objective and dynamics of DEP you explore in these experiments is pretty much equivalent to what is used in the DL, the DEP theoretical framework indeed seems to be more general.
> > 2. You have mostly dispelled my concerns, but I have a follow-up question regarding AEP and the $\beta \rightarrow 0$ limit. I agree that if x is initialized as x0 then the correction term vanishes. What happens if this is not exactly the case and x is initialized slightly away from x0? Will x then gradually move towards or away from x0?
> > 3. The stability experiments in the rebuttal for FTzu are interesting but given the low accuracies I assume it is a tiny network with a single hidden layer? I would expect oscillations to be a bigger concern in networks with multiple hidden layers. Did you do experiments with asymmetric (or even anti-symmetric) weights where the feedback weights are not frozen? (The experiments in the paper were with just one hidden layer and as far as I can tell the new CIFAR10 experiments are feedforward (so static 0-valued feedback weights), and the new Fashion-MNIST results have frozen but non-zero feedback weights.)

---

> > > ### Author Response · Authors · 2026-04-06
> > >
> > > We thank the reviewer for recognizing the broader scope of our DEP framework, and for their follow-up questions.
> > >
> > > ### Free Equilibrium Convergence
> > >
> > > At $\beta = 0$, the augmented dynamics are:
> > >
> > > $$\frac{dx}{dt} = F(x, \theta) - 2A_{\mathcal{J}}(\overline{x}^0, \theta)(x - \overline{x}^0)$$
> > >
> > > Introducing a perturbation $x(t) = \overline{x}^0 + \delta x(t)$ and linearizing $F(x, \theta)$ around the free equilibrium (where $F(\overline{x}^0, \theta) = 0$), the evolution of the perturbation is:
> > >
> > > $$\frac{d(\delta x)}{dt} \approx \mathcal{J}\_F(\overline{x}^0, \theta)\delta x - 2A\_{\mathcal{J}}(\overline{x}^0, \theta)\delta x$$
> > >
> > > Using the standard decomposition in symmetric and antisymmetric part, as done in the manuscript, $\mathcal{J}_F (\overline{x}^0, \theta) = S\_{\mathcal{J}} + A\_{\mathcal{J}}$, the effective Jacobian simplifies to the transpose of the original Jacobian:
> > >
> > > $$\frac{d(\delta x)}{dt} = (S_{\mathcal{J}} + A_{\mathcal{J}} - 2A_{\mathcal{J}})\delta x = (S_{\mathcal{J}} - A_{\mathcal{J}})\delta x = \mathcal{J}_F^\top \delta x$$
> > >
> > > Because the free phase successfully converged to $\overline{x}^0$, the eigenvalues of the original Jacobian $\mathcal{J}\_F(\overline{x}^0, \theta)$ must have strictly negative real parts. Since a square matrix and its transpose share the exact same eigenvalues, the augmented system defined by $\mathcal{J}_F^\top(\overline{x}^0, \theta)$ is also locally asymptotically stable. The perturbation $\delta x$ will exponentially decay, and $x$ will converge back to $\overline{x}^0$. Thus $\overline{x}^0$ will be a stable attractor also in the augmented system. Notably, because this analysis is local, initializing the system far from the free equilibrium in the nudged phase may cause it to converge to spurious equilibria. We emphasize, however, that this is an inherent limitation of EP.
> > >
> > > ---
> > >
> > > **Oscillations**
> > >
> > > We thank the reviewer for this observation. Due to time constraints during the rebuttal, our initial stability experiments indeed used a small network. To address your concern regarding oscillations in more complex architectures, we scaled the experiment to a non-layered network with 500 hidden neurons using fully recurrent, all-to-all connectivity (excluding direct input-to-output connections). After adapting hyperparameters, our new results show that the network continues to improve stability at the end of training. The results are shown in the table below.
> > >
> > > **Table 1:** Output variance and final test accuracy on MNIST (%) across different values of $r_{in}$ with $r_{str} = 0.7$. (mean $\pm$ std over 10 runs) (500 hiddens, all-to-all, no input-output).
> > >
> > > | **$r_{in}$** | **Output variance (Untrained)** | **Output variance (Epoch 80)** | **Test Acc. (%) (Epoch 80)** |
> > > | :--- | :--- | :--- | :--- |
> > > | **0.01** | $(3.38 \pm 0.90) \times 10^{-4}$ | $(5.22 \pm 2.34) \times 10^{-5}$ | $36.34 \pm 6.25$ |
> > > | **0.10** | $(2.33 \pm 0.48) \times 10^{-4}$ | $(1.39 \pm 0.17) \times 10^{-4}$ | $90.54 \pm 0.19$ |
> > > | **0.50** | $(1.34 \pm 0.32) \times 10^{-5}$ | $(1.06 \pm 0.25) \times 10^{-6}$ | $94.96 \pm 0.10$ |
> > > | **1.00** | $(6.27 \pm 1.24) \times 10^{-7}$ | $(1.75 \pm 0.50) \times 10^{-8}$ | $96.30 \pm 0.09$ |
> > >
> > > As for EP, the stability analysis will depend on the considered substrate. While we do not expect all systems to be stable, for those that do, we provide mathematical guarantees that AEP will yield the true gradient when $\beta \rightarrow 0$.
> > >
> > > ---
> > >
> > > **Frozen Backward Connections**
> > >
> > > We wish to clarify that in the revised manuscript, we extend the first experimental part to Fashion-MNIST using a network where both forward and backward weights are learned (neither is fixed). With the right choice of hyperparameters (sufficiently small Euler steps), learning is stable, and we achieve $86.41 \pm 0.22\%$ on a 500-500 bidirectional asymmetric network. Preliminary results on a 500-500-500 network achieve 86.7% on Fashion-MNIST.

---

### Official Review · Reviewer_Q3zH · 2026-03-22

**Soundness:** 2
**Presentation:** 3
**Significance:** 3
**Originality:** 3
**Overall Recommendation:** 4
**Confidence:** 3

**Summary:**

This paper addresses a limitation of the standard Equilibrium Propagation (EP) algorithm: its restriction to conservative systems. Identifying the mathematical shortcoming of previous extensions to non-conservative systems, specifically, the Vector Field (VF) algorithm's inability to compute the exact gradient of the cost function, this paper proposes an extended EP framework applicable to arbitrary non-conservative systems. This proposed framework consists of two mathematically equivalent approaches:

- Asymmetric EP (AEP): This approach modifies the learning dynamics by introducing a local corrective force into the system's dynamics during the nudged phase.
- Dyadic EP: A variational approach that maps non-reciprocal dynamics onto an energy landscape, thereby enabling exact gradient computation.

Experiments on the MNIST dataset show that AEP successfully trains asymmetric continuous Hopfield networks and feedforward networks. The proposed method achieves better accuracy and faster learning speeds than traditional EP and VF. These improvements are especially clear in configurations with strong structural asymmetry.

**Compliance With Llm Reviewing Policy:**

Affirmed.

**Final Justification:**

The rebuttal and CIFAR-10 results demonstrate the framework's potential. However, hardware mechanisms for decoupling and asymmetry extraction face significant engineering hurdles. Given the solid mathematical rigor and theoretical value, I maintain my score of 4 (Weak Accept).

**Key Questions For Authors:**

- As mentioned in **W1**, the current experiments are limited to the MNIST dataset. Can the authors provide additional results on more complex datasets (such as CIFAR-10) or deeper network architectures? If this is not currently possible, what is the main bottleneck that prevents the algorithm from being applied to more complex tasks?

- As mentioned in **W2**, training feedforward networks with AEP implicitly requires that the hardware implementation supports the physical activation of the backward pathways. Does this contradict the original goal of the EP algorithm, which is to avoid explicit backward physical circuits? How do the authors plan to justify or solve this issue in actual hardware implementations?

**Limitations:**

Yes.

**Strengths And Weaknesses:**

**Strengths**
- **Theoretical novelty and contribution:** The paper addresses a long-standing limitation of the standard EP algorithm, which was previously restricted to conservative systems. The proposed framework provides a theoretical foundation for applying physics-inspired learning to a wider range of asymmetric feedforward networks.
- **Mathematical rigor:** The mathematical derivations are strict and solid. The authors provide a rigorous derivation of the gradient error in the previous VF algorithm. Furthermore, they mathematically prove that the gradients computed by their proposed framework are equivalent to those calculated by BPTT.

**Weaknesses**
- **(W1) The experimental evaluation is limited:** All validations are performed on the MNIST dataset. For a conference like ICML, validation on more complex tasks or deeper networks is expected.
- **(W2) The proposed method seems to partially undermine the original advantage of the EP algorithm:** To train feedforward networks with AEP, the algorithm implicitly requires the hardware to physically support and activate backward connections. This requirement seems to reduce the practical value of EP.

---

> ### Author Rebuttal · Authors · 2026-03-31
>
> ### W1 & Q1: Experimental Scale (MNIST vs. Fashion-MNIST, CIFAR-10 / Deeper Networks)
>
> We thank the reviewer for emphasizing scalability. To address this, we have updated the 3 (now 4) parts of our numerical experiments:
> * Part 5.1: Scaled to deeper networks on Fashion-MNIST (see this review).
> * Part 5.2: Added supplementary experiments evaluating stability and oscillation in Appendix (see Reviewer FTzu).
> * Part 5.3: Extended to compare different architectures and inference dynamics on Fashion-MNIST (see Reviewer USLV).
> * Part 5.4 (New): Evaluated AEP, DEP, VF, and BP on CIFAR-10 (see this review).
>
> The appendix has been updated as well as the architectural details, experimental setups, and new results.
>
> First, we updated the experiments to directly compare Vector Field (VF), Asymmetric Equilibrium Propagation (AEP), and Equilibrium Propagation (EP) with symmetric initialization on Fashion-MNIST using a deeper bidirectional network of two hidden layers of 500 and 200 neurons, presented alongside the MNIST results. We observe similar performance improvements of AEP.
>
> Then, we added new experiments on the more complex **CIFAR-10** dataset using a deeper, feedforward, 9-layer CNN (**CNN9**). Specifically, CNN9 is a VGG-style architecture comprising 8 convolutional layers (with channel dimensions expanding from 64 to 512) and 1 final linear classifier.
>
> Table 1 shows Dyadic Equilibrium Propagation (DEP) and AEP scale successfully. DEP achieves 90.68% accuracy, nearly matching Backpropagation (BP) at 40 epochs. Moreover, this experiment highlights standard VF's primary bottleneck in deep and feedforward architectures as we are not able to successfully address the task, facing instability issues. DEP and AEP explicitly resolve this (we refer to reviewer FTzu for a discussion of the differences between these two algorithms).
>
> | **Training Method** | **Test Acc. (7 Epochs)** | **Test Acc. (40 Epochs)** |
> | :--- | :---: | :---: |
> | BP | 75.84% | 90.87% |
> | DEP | 76.60% | 90.68% |
> | AEP | 74.38% | 89.70% |
> | VF | 29.48% | 10.00% |
>
> *Table 1: CIFAR-10 test accuracies after 7 and 40 epochs using the CNN9 architecture.*
>
> We once again thank all the reviewers for requesting these additional experiments. While our initial submission focused on the MNIST dataset as a proof-of-concept, the inclusion of these Fashion-MNIST and CIFAR-10 results have allowed us to more effectively demonstrate the true value and robustness of our proposed methods.
>
> ### W2 & Q2: AEP Hardware feasibility
>
> We thank the reviewer for this question. AEP does not contradict EP's original goals. EP's essence isn't the absence of backward communication—it assumes symmetric weights $(W_{ij}=W_{ji})$, necessitating physical backward pathways. Its defining feature is realizing inference and learning via continuous relaxation of a single physical system, unlike backpropagation, which requires a distinct, non-physical secondary circuit to compute and route error signals.
>
> AEP fully preserves this essential relaxation mechanism. It is the exact same physical system to which we add an **additive, local corrective force** during the nudged phase. This mirrors the nudging mechanism; however, instead of merely nudging toward an output, we push the system toward the correct equilibrium to learn non-conservative forces.
>
> We tested feedforward networks to show AEP succeeds where standard EP and VF fail. While AEP requires backward connections, any physical learning algorithm for feedforward networks needs backward pathways during training to transmit error signals (hence VF's collapse). In hardware, this implies designing feedback lines active during training, but decoupled during inference for energy efficiency.
>
> Finally, regarding hardware feasibility: algorithmic implementation is always substrate-dependent. Just as standard EP only yields local learning rules for specific energy functions (*e.g.*, quadratic interactions), AEP requires a physical substrate capable of extracting the local asymmetry $A(\bar{x}^0, \theta)$ at the synapse level. Finding the exact hardware mechanism to implement this correction is the subject of ongoing work. A promising insight is that existing methods, such as Phaseless Alignment Learning (Max et al, 2024), can already align weights locally to make a system more symmetric. These methods implicitly utilize information about the system's asymmetry, which is precisely the information required to implement or approximate our AEP correction.

---

> > ### Author Rebuttal · Reviewer_Q3zH · 2026-04-03
> >
> > I thank the authors for the detailed explanation and the CIFAR-10 results. Regarding W2, the authors provided a helpful clarification on the conceptual alignment between AEP and EP. However, the practical implementation of mechanisms such as decoupling feedback lines and extracting local asymmetry appears to face significant engineering hurdles on current hardware substrates. Additionally, as Reviewer KfVN noted, the use of frozen weights in deeper architectures suggests that the stability and scalability of learning in such systems could benefit from more extensive validation. Given the mathematical rigor and the theoretical value of extending the EP framework to non-conservative systems, I maintain my positive assessment and keep my score of 4 (Weak Accept).

---

> > > ### Author Response · Authors · 2026-04-08
> > >
> > > Thank you for your continued engagement. Regarding the numerical experiments, the revised manuscript now extends the first experiment to Fashion-MNIST using a network where both forward and backward weights are learned. By choosing sufficiently small Euler steps for inference, learning remains stable. We achieve $86.41 \pm 0.22$% on a 500-500 bidirectional asymmetric network, and our preliminary results on a 500-500-500 network reach $86.7$%. We also added supplementary stability results in answer to reviewer KfVN (Oscillations part).

---

### Decision · Program_Chairs · 2026-04-30

**Decision:**

Accept (regular)

**Comment:**

This paper extends the Equilibrium Propagation method, previously proposed for conservative systems, to non-conservative systems. The primary concern was insufficient experiments. While this concern cannot be said to have been fully addressed, it has been mitigated to some extent by the authors' explanations. On the other hand, the reviewers acknowledge that this method is a valuable contribution based on a strong theoretical foundation. Therefore, this paper should be accepted.